# TIVIT: TIME SERIES REPRESENTATIONS LIE HIDDEN IN PRETRAINED VISION TRANSFORMERS

## ABSTRACT

Time series classification is a fundamental task in healthcare and industry, yet the development of time series foundation models (TSFMs) remains limited by the scarcity of publicly available time series datasets. In this work, we propose **Ti**me **Vi**sion **T**ransformer (**TiViT**), a framework that converts time series into images to leverage the representational power of frozen Vision Transformers (ViTs) pretrained on large-scale image datasets. TiViT achieves state-of-the-art performance on time series classification and anomaly detection benchmarks by utilizing the hidden representations of large OpenCLIP models. We explore the structure of TiViT representations and find that intermediate layers with high intrinsic dimension are the most effective for time series classification. Furthermore, we assess the alignment between TiViT and TSFM representation spaces and identify a strong complementarity, with additional performance gains achieved by combining their features. Finally, we provide theoretical and qualitative insights about the benefits of 2D patching for time series modeling with ViTs. Our findings reveal a new direction for reusing vision representations in a non-visual domain.

## 1 INTRODUCTION

Foundation models have disrupted the field of machine learning. Typically built upon the Transformer (Vaswani et al., 2017) architecture, they are trained on large-scale datasets to learn generalizable representations for a wide range of downstream tasks. Vision models like DINOv3 (Siméoni et al., 2025) can be applied in image classification or segmentation with minimal supervision. Vision language models (VLMs) such as CLIP (Radford et al., 2021) or SigLIP (Tschannen et al., 2025; Zhai et al., 2023) can even be transferred to new tasks without any supervision since they have learned to ground semantic concepts in natural language. VLMs have been increasingly applied in new domains, including audio (Dixit et al., 2024; Xie et al., 2024) and medicine (Zhang et al., 2024).

Time series capture critical information in healthcare, finance, and manufacturing. Inspired by the success of foundation models in natural language processing (NLP) and computer vision (CV), similar models have recently been developed for the analysis of time series, following two different approaches. The first one is to pretrain time series foundation models (TSFMs) in a self-supervised way (Ansari et al., 2024; Das et al., 2024; Feofanov et al., 2025; Goswami et al., 2024; Lin et al., 2023) using a large-scale real-world time series dataset. The second one is to repurpose foundation models from other domains, such as NLP (Jin et al., 2024; Zhou et al., 2023) and CV (Chen et al., 2024; Li et al., 2023b), for time series tasks. The idea behind this approach is to benefit from the vast amount of samples that large vision and language models are trained on, and which are often unavailable in the time series domain.

Adapting vision models to time series analysis is particularly compelling, since time series can be visualized as line plots, heatmaps, or spectrograms (Ni et al., 2025). TimesNet (Wu et al., 2023) has been trained end-to-end on heatmaps generated from time series, pretrained Masked Autoencoders have been applied in zero-shot time series forecasting (Chen et al., 2024), and pretrained SwinTransformers have been finetuned on line plots of irregularly sampled time series (Li et al., 2023b). However, these approaches are either restricted to the task of forecasting or require costly per-dataset training and finetuning. No prior work has shown that large vision and vision-language models trained on billions of images can be state-of-the-art in time series classification. Furthermore, there is no theoretical explanation yet for the effectiveness of 2D time series modeling.

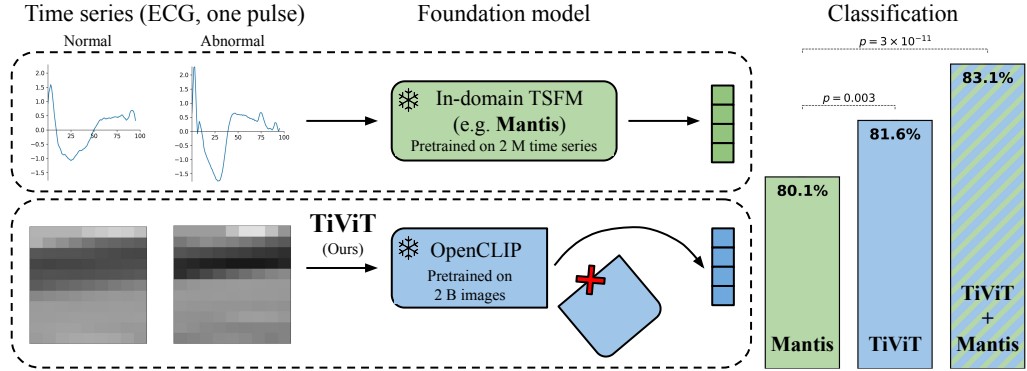

Figure 1: While TSFMs operate directly on the 1D time series signal, TiViT transforms time series into images to leverage pretrained ViTs for feature extraction. We display the average time series signal of two classes from ECG200 (Olszewski, 2001) and their corresponding 2D representations. Utilizing the hidden representations of OpenCLIP, TiViT significantly outperforms Mantis in linear classification on the UCR benchmark. Combining both models further improves accuracy.

In this work, we demonstrate that frozen vision foundation models such as OpenCLIP, SigLIP 2, and DINOv3, pretrained solely on natural images or image-text pairs, can serve as universal feature extractors for time series tasks without any pretraining or finetuning on time series data. The transformation of time series into images for feature extraction with ViTs is motivated by the intuition that 2D modeling spatially distributes label-relevant information across patches, thereby facilitating classification. To validate this hypothesis, we investigate 1D and 2D time series modeling with Transformers, offering insights into when and why image-based modeling can be advantageous.

Our main contributions are summarized as follows: (1) We introduce the Time Vision Transformer (TiViT), leveraging hidden representations of pretrained and frozen ViTs for time series analysis. TiViT surpasses conventional TSFMs without any fine-tuning in time series classification across 128 datasets and time series anomaly detection across 248 datasets. (2) We study the alignment of TiViTs and TSFMs and find that they extract complementary information from time series. By merging their representations, we achieve an average improvement of +3% over TSFMs in time series classification. (3) We provide a theoretical and empirical analysis at the patch level of Transformers, showing that the image-based modeling of time series reduces sample complexity and thus makes training more efficient than conventional 1D modeling.

## 2 RELATED WORK

**Time series foundation models** Recently, the research community has witnessed an impressive surge in the number and variety of TSFMs. At first, such models were based on repurposing large language models (LLMs) for time series tasks (Cao et al., 2024; Chang et al., 2025; Gruver et al., 2023; Jin et al., 2024; Xue & Salim, 2024; Zhou et al., 2023), exploiting the ability of LLMs to efficiently handle sequential data. A different approach that gained in popularity later was to train TSFMs from the ground up on extensive and diverse datasets (Ansari et al., 2024; Bhethanabhotla et al., 2024; Das et al., 2024; Feofanov et al., 2025; Gao et al., 2024; Goswami et al., 2024; Lin et al., 2023; Liu et al., 2024a;b; Rasul et al., 2024; Wang et al., 2024). While most of the models were designed for time series forecasting, several of them also specifically tackled time series classification (Feofanov et al., 2025; Gao et al., 2024; Goswami et al., 2024; Lin et al., 2023; Zhou et al., 2023). These TSFMs are on par with or exceed the performance of earlier deep learning models such as the famous TimesNet (Wu et al., 2023), which has been trained separately per dataset.

**Transforming time series into images** Time series can be transformed into images in many ways, either based on the 1D representation of the time series in the original space (line plot) or frequency space (spectrogram), or by using a 2D modeling approach (heatmap, Gramian angular field, recurrence plot) that stacks segments of the input time series based on a chosen periodicity. Vision models, often based on CNNs and their variations, were applied on such image-based representa-

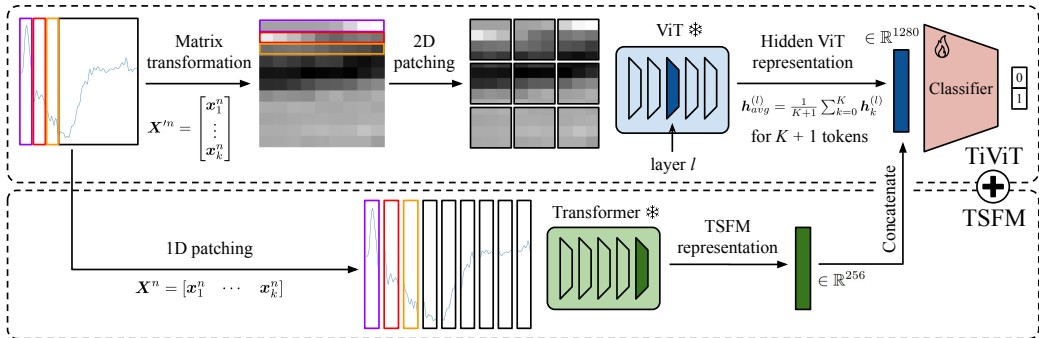

Figure 2: Illustration of TiViT on a time series sample from ECG200 (Olszewski, 2001). We split the time series into segments and stack them to form a grayscale image. Then, we patch the image in 2D and feed it into a frozen ViT pretrained on large-scale image datasets. We average the hidden representations from a specific layer and pass them to a learnable classification head. Concatenating the representations of TiViT and TSFMs prior to classification improves accuracy.

tions of time series as early as 2013 (see Ni et al. (2025) for a recent survey). Most of them, however, were trained in a supervised way to fit the dataset at hand. This work explores how pretrained vision models can be used as powerful feature extractors without training or fine-tuning. Li et al. (2023b) showed that pretrained ViTs can be effective in the classification of irregular time series from their line plot representations after full finetuning. In a similar vein, Chen et al. (2024) applied a masked auto-encoder with a pretrained frozen ViT to 2D transformed time series to perform time series forecasting. Different from these works, our TiViT model surpasses the performance of frontier TSFMs across a broad set of classification and anomaly detection benchmarks. Moreover, we explain why modeling time series in 2D rather than 1D can benefit time series classification with Transformers.

## 3  TIVIT: TIME SERIES CLASSIFICATION WITH PRETRAINED VITS

We introduce the Time Vision Transformer (TiViT) leveraging pretrained frozen ViTs from the vision or vision-language domain for time series classification. We consider a multivariate time series dataset $\mathcal{T} = \{t^n | t^n \in \mathbb{R}^{T \times D}\}_{n=1}^{N}$ containing $N$ samples, each of length $T$ and dimensionality $D$. The corresponding targets $\mathcal{Y} = \{y^n\}_{n=1}^{N}$ are labels $y^n \in \{1, ..., C\}$ from $C$ different classes. We transform the time series into images and apply ViTs on these images to extract representations for linear classification. Figure 2 illustrates our approach.

**Time series-to-image transformation**  Following the channel independence assumption, proposed by Nie et al. (2023) and widely adopted in most recent TSFMs (Feofanov et al., 2025; Goswami et al., 2024), we first split a multivariate time series $t^n \in \mathbb{R}^{T \times D}$ into $D$ univariate time series $\{t_d^n \in \mathbb{R}^T\}_{d=1}^{D}$. We then normalize each univariate time series $t_d^n$ using robust scaling, defined as: $\frac{t_d^n - Q_2}{Q_3 - Q_1}$, where $Q_1, Q_2, Q_3$ are the first, second (median), and third quartiles, respectively. We apply padding at the beginning of each time series by replicating its first value and subsequently segment it into $M$ patches $\{x_m\}_{m=1}^{M}$ of size $P$. Given a patch length $P$ and stride $S$, the total number of patches is: $M = \lfloor \frac{T-P}{S} \rfloor + 1$. We stack the patches to generate a 2D representation $X' \in \mathbb{R}^{M \times P}$, which we then render into a grayscale image $X' \in \mathbb{R}^{M \times P \times 3}$ by replicating its signals across three channels. To align with the square input resolution $(R, R)$ expected by the ViT, we resize the image.

**Time series classification**  We feed each grayscale image $X'$ representing a univariate time series into a pretrained and frozen ViT $v$ with $L$ hidden layers. The ViT inherent 2D patching yields a sequence $\{x_k' \in \mathbb{R}^{U^2}\}_{k=1}^{K}$ of flattened patches where $(U, U)$ is the resolution per patch and $K = R^2/U^2$ is the resulting number of patches. ViTs generally prepend a classification token to this sequence. The ViT consumes all input tokens and produces a sequence of features at every layer: $v(X') = \left\{[h_0^{(l)}, h_1^{(l)}, ..., h_K^{(l)}]\right\}_{l=0}^{L}$. To obtain a single embedding vector $e$ per image, we select a specific layer $l$ and average its $K + 1$ representations: $e = h_{avg}^{(l)} = \frac{1}{K+1}\sum_{k=0}^{K} h_k^{(l)}$. For

multivariate time series, we feed per-channel image representations $\{\boldsymbol{X}'_d\}_{d=1}^D$ separately into the ViT and concatenate the resulting embeddings for a specified layer: $\text{Concat}(\boldsymbol{e}_1, ..., \boldsymbol{e}_D)$. We only train a linear classifier on the ViT representations and their corresponding class labels. To enhance performance, the embeddings of frozen TSFMs and ViTs can be concatenated prior to classification.

# 4 EXPERIMENTAL EVALUATION

In this section, we evaluate TiViT on the discriminative time series tasks of classification and anomaly detection, showing its state-of-the-art performance compared to supervised, self-supervised, and FM competitors. Although the contrastive pre-training ViTs rely on is not suitable for generative tasks, we further provide promising preliminary results with TiViT in long-term multivariate forecasting. We note that classification and forecasting are two fundamentally different tasks, and most existing TSFMs concentrate exclusively on only one of them.

For classification, we evaluate TiViT on 128 univariate time series dataset from the UCR benchmark (Dau et al., 2019) and on 27 multivariate datasets from the UEA benchmark (Bagnall et al., 2018). Our study examines three differently pretrained ViTs: OpenCLIP (Cherti et al., 2023; Ilharco et al., 2021), SigLIP 2 (Tschannen et al., 2025), and DINOv3 (Siméoni et al., 2025). We compare TiViT to the state-of-the-art TSFMs Mantis (Feofanov et al., 2025) and Moment (Goswami et al., 2024) which are exclusively pretrained on time series. We further consider GPT4TS (Zhou et al., 2023) pretrained on textual data, the forecasting TSFMs VisionTS (Chen et al., 2024) and Chronos (Ansari et al., 2024), and a wide range of (self-)supervised baselines (pre-)trained per time series dataset. To evaluate the effectiveness of TiViT and TSFM representations in time series classification, we train a logistic regressor with the LBFGS solver per dataset. A detailed overview of our experimental setup is provided in Appendix C.

## 4.1 TRANSFORMING TIME SERIES INTO IMAGES FOR FEATURE EXTRACTION

The performance of our time series-to-image transformation is sensitive to the patch size $P$, as extreme values can create redundant visual tokens during resizing to the ViT input resolution (see Figure 10). To avoid a computationally expensive hyperparameter search for the optimal patch size $P^*$ per dataset, we propose the heuristic $P = \sqrt{T}$ for any series of length $T$. This choice yields a square-shaped matrix representation prior to resizing, which minimizes horizontal or vertical distortion and thus preserves patch diversity. While an exhaustive search for $P^*$ offers a marginal accuracy improvement in the case of no overlap, our heuristic provides a strong baseline at a fraction of the computational cost. As displayed in Figure 3, introducing overlap between patches further boosts performance and makes the impact of the optimal patch size vanish. Details can be found in Appendix D.1. Consequently, we use a patch size of $P = \sqrt{T}$ and a stride of $S = P/10$ in the following experiments.

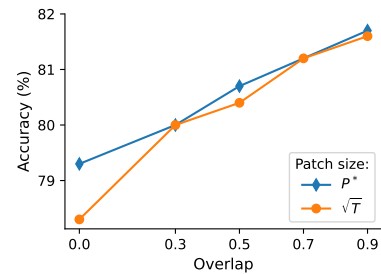

Figure 3: Effect of patch size and overlap on classification accuracy.

## 4.2 EFFECTIVENESS OF HIDDEN VIT REPRESENTATIONS

We repurpose frozen ViTs as feature extractors for time series data. While the final representations of ViTs typically capture high level semantics, intermediate layers encode lower level information (Dorszewski et al., 2025). Our study reveals that the intermediate representations of ViTs are the most effective for downstream classification. In Figure 4a we report the classification performance of TiViT with pretrained ViTs from DINOv3, CLIP, and SigLIP 2 on the validation split of the UCR benchmark. For each dataset, we extract representations from the hidden layers of ViTs, average them, and train a linear classifier. The intermediate representations of ViTs, between 40% and 70% of the layer depth, achieve the highest classification accuracy. CLIP and SigLIP 2, both optimized with a contrastive loss on image-text pairs, reach best performance in their earlier layers: layer 14 of 33 for CLIP (ViT-H) and layer 12 of 28 for SigLIP 2 (SoViT-400m). In contrast, DINOv3 (ViT-L) trained with contrastive learning and masked modeling on images only, reaches the highest

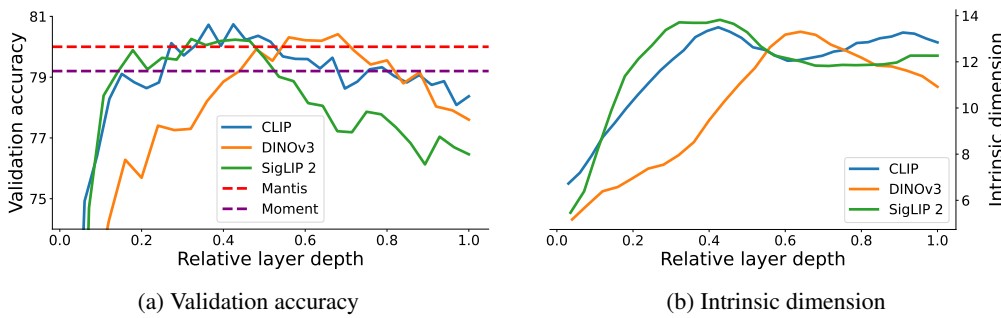

(a) Validation accuracy

(b) Intrinsic dimension

Figure 4: (a) Validation accuracy and (b) Intrinsic dimensionality using hidden representations at different depth of pretrained ViTs (CLIP, DINOv3, SigLIP 2). Results are averaged over 128 datasets from the UCR benchmark.

Table 1: Classification accuracy of TSFMs and TiViT per benchmark.

| Model | UCR | UEA |
|---|---|---|
| Moment | 79.0 | 69.9 |
| Mantis | 80.1 | 72.4 |
| TiViT *(Ours)* | 81.6 +1.5 | 72.0 -0.4 |
| TiViT + Moment *(Ours)* | 82.7 +2.6 | 72.6 +0.2 |
| TiViT + Mantis *(Ours)* | **83.1** +3.0 | **73.7** +1.3 |

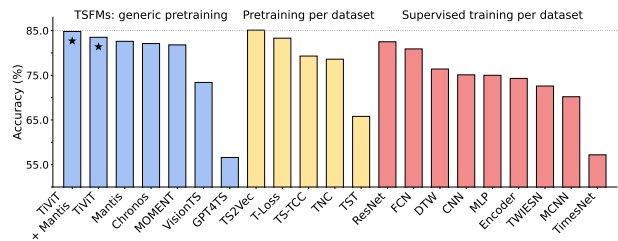

Figure 5: Classification accuracy across 91 UCR datasets. SL and SSL baselines from Goswami et al. (2024).

classification accuracy with representations from a later layer (17 of 25). For each ViT, we determine the optimal hidden layer based on its highest validation accuracy on the UCR benchmark.

**Intrinsic dimension** To better understand the hidden representations of ViTs, we analyze their intrinsic dimension (see Figure 4b) and principal components (see Appendix D.5). Valeriani et al. (2023) have previously investigated the geometry of hidden representations of Transformers for in-domain vision and language applications. We measure the intrinsic dimension of ViTs applied on time series from the UCR archive using the DADApy (Glielmo et al., 2022) implementation of the TWO-NN estimator (Facco et al., 2017). Figure 4b displays for three different ViT backbones the intrinsic dimensionality of their representations at varying layer depth. Across these three backbones, the mean Pearson correlation coefficient between the intrinsic dimension and validation accuracy is $\rho = 0.704$. The best performing layers exhibit the highest or second highest intrinsic dimension.

**Benchmark** Unless stated otherwise, we refer to our best-performing model with 14 layers of Open-CLIP ViT-H as TiViT. A full comparison of TiViT and TSFMs on the UCR and UEA test set is reported in Table 1. The state-of-the-art TSFM Mantis achieves a linear classification accuracy of 80.1% on the UCR benchmark. Our statistical analysis with a paired t-test and a significance level of 0.05 confirms that TiViT significantly outperforms ($p = 0.003$) Mantis across the 128 datasets of the UCR benchmark, achieving 81.6% accuracy. We further extend our analysis to multivariate time series. TiViT reaches a classification accuracy of 72.0%, which is statistically on par with Mantis on the UEA benchmark. The concatenation of per-channel representations, without learning any explicit cross-channel interactions, achieves state-of-the-art performance. In line with prior work (Feofanov et al., 2025), we could not observe any consistent benefit with channel-gating or attention pooling of channel-wise representations. Figure 5 shows that TiViT outperforms not only other TSFMs, but also a series of supervised learning (SL) and self-supervised learning (SSL) methods (pre-)trained per dataset. The comparison is limited to 91 UCR datasets since most of these models can only handle time series up to $T = 512$. Interestingly, TSFMs such as Chronos and VisionTS, primarily designed for forecasting, perform worse than TiViT or Mantis in time series classification. This highlights that models optimized for forecasting cannot be simply transferred to classification tasks and emphasizes the need for dedicated classification-focused TSFMs such as TiViT.

Table 2: Joint classification accuracy and alignment score for TiViTs and TSFMs on UCR.

| Fusion | Model 1 | | Model 2 | | Joint accuracy | Alignment score |
|---|---|---|---|---|---|---|
| | Name | Acc | Name | Acc | | |
| TSFM × TSFM | Mantis | 80.1 | Moment | 79.0 | 81.5 (+1.4, +2.5) | 0.222 |
| TiViT × TiViT | CLIP | 81.6 | DINOv3 | 80.2 | 82.2 (+0.6, +2.0) | **0.431** |
| TiViT × TSFM | DINOv3 | 80.0 | Moment | 79.0 | 82.0 (+2.0, +3.0) | 0.213 |
| | DINOv3 | 80.0 | Mantis | 80.1 | 82.5 (+2.5, +2.4) | 0.243 |
| | CLIP | 81.6 | Moment | 79.0 | 82.7 (+1.1, +3.7) | 0.241 |
| | CLIP | 81.6 | Mantis | 80.1 | **83.1** (+1.5, +3.0) | 0.262 |

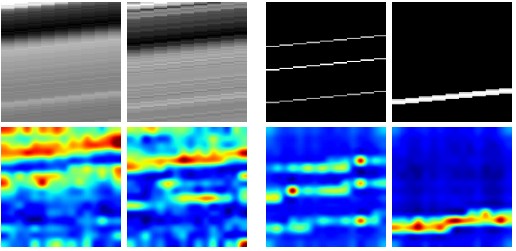

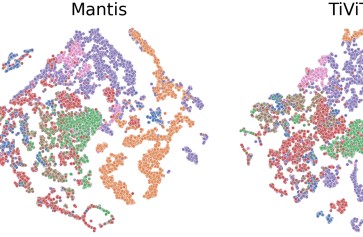

(a) Attention rollout                    (b) t-SNE visualization

Figure 6: Qualitative analysis of TiViT representations on samples from the UCR benchmark.

## 4.3 ALIGNMENT AND FUSION OF TIVIT AND TSFM REPRESENTATIONS

We do not only compare the effectiveness of TiViT and TSFM representations against each other, but also explore their complementarity when concatenating their features for joint classification. As depicted in Table 2, the combination of TiViT and TSFM consistently improves the classification performance over any standalone model. While the combination of two TSFMs yields 81.5% accuracy, fusing TiViT-CLIP with Moment and Mantis leads to even higher accuracies of 82.7% and 83.1%, respectively. These results underscore the potential of multimodal time series analysis.

To uncover the differences between TiViTs and TSFMs, we assess the alignment of their representation spaces using the mutual k-nearest neighbor metric (Huh et al., 2024) on the 10 largest UCR datasets. Table 2 presents the average alignment scores across datasets for CLIP, DINOv3, Mantis, and Moment. Interestingly, the alignment score of the two TSFMs is relatively low. We hypothesize that this discrepancy arises from their different pretraining paradigms. A similarly low alignment score is observed between any TiViT and TSFM, which we attribute to their domain gap. TiViT and Mantis extract different representations for the same time series, which is beneficial for joint classification. The highest alignment is measured between TiViT-CLIP and TiViT-DINOv3, both of which are pretrained contrastively on image datasets.

## 4.4 FEATURE VISUALIZATION

To gain insights into the processing of TiViT, we employ attention rollout (Abnar & Zuidema, 2020) on images generated from the ECG200 and ElectricDevices datasets of the UCR archive. As shown in Figure 6a, the attention weights aggregated across layers highlight the most salient regions of the input. In particular, high attention weights align with the bright and dark areas in the 2D image representation, which correspond to high and low signals in the original time series. This indicates that TiViT attends to the critical signals necessary for distinguishing samples of different classes.

In addition, we compare the representations of TiViT and Mantis in Figure 6b using t-SNE (van der Maaten & Hinton, 2008) visualizations on the ElectricDevices dataset. Mantis is trained contrastively and can discover class-distinguishable time series representations without label supervision. Our findings go beyond this: even without any training on time series, TiViT generates embeddings that form clusters aligned with the ground-truth classes illustrated in different colors. More t-SNE visualizations for TiViT on UCR datasets are provided in Appendix D.11.

Table 3: Classification accuracy on UCR subsets (left) and comparison of classifiers (right).

| Model | UCR subsets | | | | Classification head | | |
|---|---|---|---|---|---|---|---|
| | Small | Large | Short | Long | Logistic R. | Nearest C. | Random F. |
| Moment | 86.6 | 85.4 | 87.4 | 67.3 | 79.0 | 68.4 | 75.7 |
| Mantis | 87.2 | 82.6 | 88.2 | 71.4 | 80.1 | 71.2 | 77.7 |
| TiViT *(Ours)* | 90.5 | 85.4 | 87.8 | 75.6 | 81.6 | 71.9 | 77.7 |
| TiViT + Moment *(Ours)* | 90.7 | **87.2** | 88.8 | 75.7 | 82.7 | 73.6 | 79.5 |
| TiViT + Mantis *(Ours)* | **91.4** | 86.2 | **89.3** | **77.8** | **83.1** | **73.8** | **80.1** |

Table 4: Anomaly detection on 248 datasets from the UCR Anomaly Archive. We compare the performance of TiViT to baselines reported by Goswami et al. (2024).

| Metric | | TiViT | MOMENT | GPT4TS | TimesNet | Anomaly TF | DGHL | k-NN |
|---|---|---|---|---|---|---|---|---|
| | Mean | **0.746** +0.118 | 0.628 | 0.424 | 0.537 | 0.492 | 0.425 | 0.554 |
| Adj. F1 | Median | **0.985** +0.207 | 0.778 | 0.331 | 0.541 | 0.432 | 0.331 | 0.595 |
| | Std | 0.368 | 0.373 | 0.366 | 0.389 | 0.401 | 0.365 | 0.393 |
| | Mean | **0.770** +0.064 | 0.684 | 0.611 | 0.679 | 0.661 | 0.646 | 0.706 |
| VUS ROC | Median | **0.795** +0.068 | 0.692 | 0.615 | 0.692 | 0.658 | 0.635 | 0.727 |
| | Std | 0.169 | 0.146 | 0.114 | 0.141 | 0.147 | 0.137 | 0.155 |

## 4.5 ABLATION STUDIES

In Section 4.2, we report the performance of TiViT across all 128 UCR datasets. To further explore its capabilities, we now select four UCR subsets: 10 datasets with the fewest training samples ($16 \leq N_{train} \leq 20$), the most training samples ($1000 \leq N_{train} \leq 8926$), the shortest time series ($15 \leq T \leq 80$), and the longest time series ($1500 \leq T \leq 2844$). The results are displayed in Table 3. TiViT significantly outperforms Mantis on subsets with a small training set (90.5% vs. 87.2%) and long time series (75.6% vs. 71.4%). These findings demonstrate that TiViT excels in generalizing from limited training data and in modeling long-range dependencies.

Finally, we investigate the effectiveness of TiViT in zero-shot classification with a nearest centroid classifier. On the UCR benchmark, TiViT achieves a zero-shot classification accuracy of 71.9%. Our approach surpasses both Mantis (71.2%) and Moment (68.4%), highlighting the ability of TiViT to extract generalizable representations. We further merge the representations of TiViT and Mantis, reaching a state-of-the-art zero-shot accuracy of 73.8%.

## 4.6 TASKS BEYOND CLASSIFICATION

As shown above, TiViT excels in time series classification, providing rich embeddings with a very strong zero-shot performance. This prompts us to apply it to anomaly detection, too, as both tasks are of a discriminative nature. To this end, Table 4 reports the performance of TiViT in time series anomaly detection following the setup considered in Goswami et al. (2024). We compare it to foundation models and specialized methods across 248 datasets from the UCR Anomaly Archive (Wu & Keogh, 2023) and observe that TiViT equipped with an OpenCLIP ViT-B backbone and a trainable linear reconstruction head achieves an adjusted best F1 score of 0.746, substantially outperforming Moment with a score of 0.628.

TiViT is especially tailored to discriminative tasks due to the large-scale contrastive pre-training of the ViT backbone. To verify the usefulness of TiViT in generative tasks, we provide preliminary results of its evaluation in long-term time series forecasting in Table 22 in the appendix. We note that TiViT reaches linear probing performance comparable to that of Moment on the 8 standard multivariate long-term forecasting datasets (Wu et al., 2021).

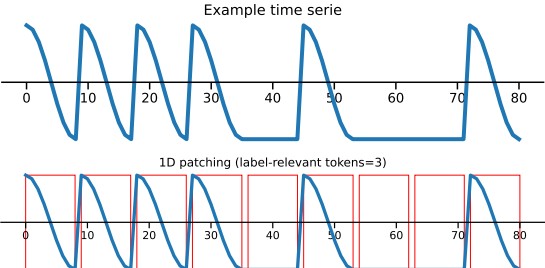 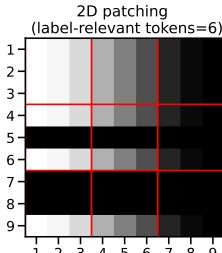

Figure 7: 2D patching yields a higher number of label-relevant tokens (with constant negative signal) than 1D patching. This facilitates time series classification with Transformers.

## 5 INSIGHTS ON MODELING TIME SERIES AS IMAGES

TiViT surpasses the performance of TSFMs in time series tasks by leveraging pretrained ViTs. This raises a key question: is its success solely due to the rich representations learned from billions of natural images, or is there an inherent advantage of the 2D patching strategy as well? We develop a theoretical insight at the patch level showing how the 2D representation of time series can enhance the classification performance of Transformer models. To empirically validate this, we compare the performance of Transformers pretrained on real-world data using 1D versus 2D patching.

### 5.1 THEORETICAL ANALYSIS OF 1D AND 2D PATCHING

We consider a binary time series classification problem with $N$ univariate training samples $\{(\boldsymbol{t}^n, y^n), y^n \in \{+1, -1\}\}_{n=1}^N$. Each time series $\boldsymbol{t}^n \in \mathbb{R}^T$ can be patched as follows:

- 1D patching: The series $\boldsymbol{t}$ is split into $k$ contiguous, non-overlapping tokens $\boldsymbol{x}_l \in \mathbb{R}^k$.
- 2D patching: The series $\boldsymbol{t}$ is reshaped into a $k \times k$ matrix, then divided into $k$ non-overlapping $\sqrt{k} \times \sqrt{k}$ patches, which are flattened to form tokens $\boldsymbol{x}'_{(i,j)} \in \mathbb{R}^k$.

This setup ensures the same number of tokens for 1D and 2D patching. Our analysis builds on the notion of label-relevant tokens introduced by Li et al. (2023a). Following their data model, we consider each token to be a noisy version of distinct patterns. In binary classification, there exist two such patterns $\{\boldsymbol{\mu}_1, \boldsymbol{\mu}_2\}, \boldsymbol{\mu}_i \in \mathbb{R}^k, \forall i$. For a time series $\boldsymbol{t}^n$ with label $y^n = 1$, tokens $\boldsymbol{x}$ that are noisy $\boldsymbol{\mu}_1$, i.e., $||\boldsymbol{x} - \boldsymbol{\mu_1}|| \leq ||\boldsymbol{x} - \boldsymbol{\mu_2}||$, are label-relevant. Similarly, for a time series $\boldsymbol{t}^n$ with label $y^n = -1$, the noisy versions of $\boldsymbol{\mu}_2$ are label-relevant.

**Benefits of 2D patching** Li et al. (2023a) showed that the sample complexity of a Transformer scales as $\mathcal{O}(1/\alpha_*^2)$ where $\alpha_*$ denotes the fraction of label-relevant tokens in the training samples. In Appendix A.2, we provide a constructive proof showing that under certain conditions, this fraction of label-relevant tokens is greater when the time series is transformed into a 2D representation compared to the conventional 1D representation. Therefore, 2D patching can lead to more efficient learning with Transformers than 1D patching. Figure 7 illustrates our idea for an exemplary time series with $T = 91$ and $k = 9$. We set $\boldsymbol{\mu}_1 = \cos(x)$ for $x \in [0, \pi]$ and define the label-relevant signal as $\boldsymbol{\mu}_2 = -1$. In the 1D case, only three tokens carry the label-relevant information, whereas in the 2D case there are six such tokens. Following Li et al. (2023a), distributing the discriminative signal across a larger number of tokens makes it easier for a Transformer to detect and leverage it.

**Interpretability scores** To confirm our hypothesis about the spread of information achieved with 2D modeling, we now illustrate it on samples from a real-world dataset from the UCR repository. In particular, we show that a model trained on 2D representations of time series has more regions that it deems relevant for predicting the class membership of the time series. To this end, we follow Early et al. (2024) and use MILLET: a framework that provides interpretability scores for timestamps within a time series given a pretrained model. For this, we train two shallow ViTs, ViT1D and ViT2D, on the BirdChicken dataset from the UCR repository. ViT1D takes as input a raw 1D time series, while ViT2D is trained on square 2D images of the time series. The only difference between

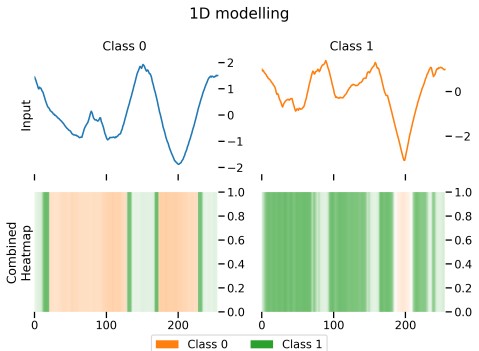 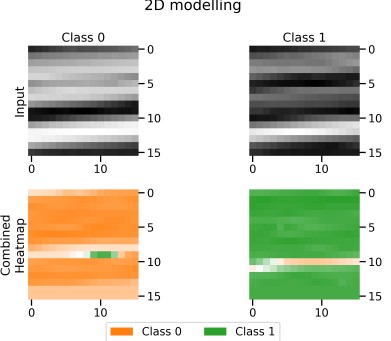

Figure 8: Comparison of interpretability heatmaps for two ViTs trained on 1D (left) and 2D (right) representations of time series from the BirdChicken dataset, respectively. The interpretability scores for the correct class of a sample are more homogeneous in the 2D case.

these two models is their patching strategy: ViT1D patches the time series using a 1D convolutional filter, while ViT2D applies a 2D convolutional filter. The obtained results for two samples from different classes are presented in Figure 8, for 1D (left) and 2D (right) cases, respectively. Note that the 1D heatmap for the sample with ground-truth class 0 highlights discriminative signals for class 1 at the beginning and end of the time series, while the corresponding 2D heatmap displays no such signals in these areas. The interpretability scores w.r.t. the ground-truth class of a sample are generally more homogeneous for ViT2D which facilitates classification.

## 5.2 PRETRAINING TRANSFORMERS WITH 1D AND 2D PATCHING

To validate our hypothesis about the benefits of 2D patching in practice, we study how patching affects the quality of representations learned by Transformers on real-world time series. In this experiment, we fix the Transformer architecture and pretraining method, and only vary the patching strategy. We then evaluate the representations learned by the model on the UCR benchmark. Following Feofanov et al. (2025), we pretrain a Transformer model with 6 layers and 8 heads

Table 5: Evaluation of models pretrained with different patching strategies on UCR.

| Patching | Non-overlap | | Overlap | |
|---|---|---|---|---|
| | 1D | 2D | 1D | 2D |
| Accuracy | 76.4 | 76.8 | 76.6 | **77.4** |

per layer using contrastive learning. Details are provided in Appendix B. We compare 1D and 2D patching with both non-overlapping and overlapping patches. As summarized in Table 5, 2D patching outperforms 1D patching, with overlapping 2D patches yielding the highest classification accuracy. This finding shows that the transformation of time series to images is not only beneficial when leveraging pretrained ViTs, but can also enhance time series pretraining from scratch.

## 6 CONCLUSION

In this paper, we introduced TiViT, the first method to successfully leverage large pretrained ViTs for time series classification. Our analysis revealed that the most effective features for this task are the hidden representations of ViTs which exhibit high intrinsic dimensionality. Building on this insight, TiViT significantly outperformed TSFMs in time series classification on the UCR benchmark and reached competitive results on UEA. We investigated the complementarity of TiViT and TSFMs, and by combining their representations, established the new state-of-the-art in zero-shot and linear classification on both benchmarks. Beyond the task of classification, TiViT excelled in time series anomaly detection on the UCR Anomaly Archive. We finally provided theoretical and empirical evidence that modeling time series in 2D rather than 1D is not only key to exploiting pretrained ViTs but broadly advantageous for time series pretraining and classification with Transformers.

**Limitations and future work** While our study evaluated time series representations via linear probing, future work could explore the finetuning of TiViT. Moreover, the powerful representations of large-scale ViTs present an opportunity for knowledge distillation into efficient time series models.

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

# A  DETAILS ON THE THEORETICAL ANALYSIS

We first review the shallow ViT and data model introduced by Li et al. (2023a) in their theoretical analysis of training a ViT. Their Theorem A.1 shows that the sample complexity for ViTs to achieve a zero generalization error is inversely correlated with the fraction of label-relevant tokens. Building on this insight, we introduce and proof Proposition 1, showing that 2D patching can increase the number of label-relevant tokens compared to 1D patching. We further illustrate our Proposition 1 with various examples of time series and their corresponding 2D representations.

## A.1  BACKGROUND

**Model and setup**   Following the setup of Li et al. (2023a), we study a binary classification problem with $N$ training samples $\{(\boldsymbol{X}^n, y^n)\}_{n=1}^N$. Each input $\boldsymbol{X}^n \in \mathbb{R}^{d \times L}$ contains $L$ tokens $\{\boldsymbol{x}_1^n, \ldots, \boldsymbol{x}_L^n\}$. Labels $y^n \in \{\pm 1\}$ are determined by majority vote over discriminative tokens. A simplified Vision Transformer (ViT) (Dosovitskiy et al., 2021) model is defined as:

$$F(\boldsymbol{X}^n) = \frac{1}{|\mathcal{S}^n|} \sum_{l \in \mathcal{S}^n} \boldsymbol{a}_{(l)}^\top \mathrm{ReLU}\left(\boldsymbol{W}_O \boldsymbol{W}_V \boldsymbol{X}^n \mathrm{softmax}\left(\boldsymbol{X}^{n\top} \boldsymbol{W}_K^\top \boldsymbol{W}_Q \boldsymbol{x}_l^n\right)\right),$$

where $\psi = (\boldsymbol{A} = \{\boldsymbol{a}_{(l)}\}_l, \boldsymbol{W}_O, \boldsymbol{W}_V, \boldsymbol{W}_K, \boldsymbol{W}_Q)$ are trainable parameters. The empirical risk minimization problem is:

$$\min_\psi f_N(\psi) = \frac{1}{N} \sum_{n=1}^N \max\{1 - y^n \cdot F(\boldsymbol{X}^n), 0\}.$$

Training uses mini-batch SGD with fixed output layer weights $\boldsymbol{A}$, following standard NTK initialization practices.

**Data model**   Tokens $\boldsymbol{x}_l^n$ are noisy versions of $M$ patterns $\{\boldsymbol{\mu}_1, \ldots, \boldsymbol{\mu}_M\}$, where $\boldsymbol{\mu}_1, \boldsymbol{\mu}_2$ are discriminative. Label $y^n$ depends on majority vote over tokens closest to $\boldsymbol{\mu}_1/\boldsymbol{\mu}_2$. Noise level $\tau$ satisfies $\tau < \kappa/4$, with $\kappa - 4\tau = \Theta(1)$.

**Generalization of ViT**   We now recap the main results from Li et al. (2023a) from which we derive our result, along with the main notations in Table 6.

**Assumption** (Initial Model Conditions, Li et al. (2023a)). *Initial weights $\boldsymbol{W}_V^{(0)}, \boldsymbol{W}_K^{(0)}, \boldsymbol{W}_Q^{(0)}$ satisfy:*

$$\|\boldsymbol{W}_V^{(0)} \boldsymbol{\mu}_j - \boldsymbol{p}_j\| \le \sigma, \quad \|\boldsymbol{W}_K^{(0)} \boldsymbol{\mu}_j - \boldsymbol{q}_j\| \le \delta, \quad \|\boldsymbol{W}_Q^{(0)} \boldsymbol{\mu}_j - \boldsymbol{r}_j\| \le \delta,$$

*for orthonormal bases $\mathcal{P}, \mathcal{Q}, \mathcal{R}$ and $\sigma = O(1/M), \delta < 1/2$.*

**Theorem** (Generalization of ViT, Li et al. (2023a)). *Under Assumption 1, with sufficient model width $m \gtrsim \epsilon^{-2} M^2 \log N$, fraction*

$$\alpha_* \ge \alpha_\# / (\epsilon_S e^{-(\delta + \tau)}(1 - (\sigma + \tau)),$$

*and sample size*

$$N \ge \Omega\left((\alpha_* - c'(1 - \zeta) - c''(\sigma + \tau))^{-2}\right),$$

*SGD achieves zero generalization error after*

$$T = \Theta\left(\frac{1}{(1 - \epsilon - (\sigma + \tau)M/\pi)\eta \alpha_*}\right)$$

*iterations.*

**Proposition** (Generalization without Self-Attention, Li et al. (2023a)). *Without self-attention, achieving zero error requires $N \ge \Omega\left((\alpha_*(\alpha_* - \sigma - \tau))^{-2}\right)$, demonstrating ViT's sample complexity reduction by $1/\alpha_*^2$.*

Table 6: Key Notations

| Notation | Description |
|---|---|
| $\alpha_*$ | Fraction of label-relevant tokens |
| $\sigma, \delta, \tau$ | Initialization/token noise parameters |
| $\kappa$ | Minimum pattern distance |
| $M$ | Total number of patterns |

## A.2 PROOF OF LABEL RELEVANCE IN 2D PATCHES

We introduce Proposition 1 that formalizes our theoretical analysis of 1D and 2D patching from Section 5.1 and provide a detailed proof.

**Proposition 1.** *For an arbitrary $\boldsymbol{\mu}_1, \boldsymbol{\mu}_2 \in \mathbb{R}^k$, let $\boldsymbol{t} = [\boldsymbol{x}_1 \ \boldsymbol{x}_2 \ \cdots \ \boldsymbol{x}_k]^\top \in \mathbb{R}^T$ where $\forall i \in [k], \boldsymbol{x}_i \in \mathbb{R}^k$ and either $\boldsymbol{x}_i = \boldsymbol{\mu}_1$ or $\boldsymbol{x}_i = \boldsymbol{\mu}_2$ with $\boldsymbol{\mu}_2$ being a label-relevant pattern. Let $|\{i : \boldsymbol{x}_i = \boldsymbol{\mu}_2\}| = n'$ and assume that $2\boldsymbol{x}' \cdot (\boldsymbol{\mu}_1 - \boldsymbol{\mu}_2) \leq \|\boldsymbol{\mu}_1\|^2 - \|\boldsymbol{\mu}_2\|^2$ whenever $|\{i : x'_i \in \boldsymbol{\mu}_2\}| \geq \sqrt{k}$. Then, it holds that*

$$\alpha_*^{2D} \geq \alpha_*^{1D} = \frac{n'}{k},$$

*and the inequality is strict if $n' \bmod \sqrt{k} > 0$.*

*Proof.* For a token $\boldsymbol{x}'^n$ to be label-relevant (aligned with $\boldsymbol{\mu}_2$), it must satisfy:

$$\|\boldsymbol{x}'^n - \boldsymbol{\mu}_2\| \leq \|\boldsymbol{x}'^n - \boldsymbol{\mu}_1\|.$$

Expanding both sides, we have that:

$$\|\boldsymbol{x}'^n\|^2 + 2\boldsymbol{x}'^n \cdot \boldsymbol{\mu}_1 + \|\boldsymbol{\mu}_1\|^2 \leq \|\boldsymbol{x}'^n\|^2 - 2\boldsymbol{x}'^n \cdot \boldsymbol{\mu}_2 + \|\boldsymbol{\mu}_2\|^2.$$

Regrouping the terms gives us the desired condition:

$$2\boldsymbol{x}'^n \cdot (\boldsymbol{\mu}_1 - \boldsymbol{\mu}_2) \leq \|\boldsymbol{\mu}_1\|^2 - \|\boldsymbol{\mu}_2\|^2. \tag{1}$$

Recall that $n'$ denotes the number of segments of $\boldsymbol{\mu}_2$ in time series $\boldsymbol{t}$. Each such segment spans $\sqrt{k}$ tokens, contributing at least $\sqrt{k}$ elements to each of them. Under the assumption of the proposition, it implies (1) and makes each of these $\sqrt{k}$ tokens label-relevant.

We now need to carefully consider how the $\boldsymbol{\mu}_2$ segments can be placed within $\boldsymbol{t}$ to understand how many tokens become label-relevant thanks to each $\boldsymbol{\mu}_2$. We consider two cases: 1) $n' = c\sqrt{k}$ for some $c \in \mathbb{N}$ satisfying $n' \in (0, k]$, and 2) $n' = c\sqrt{k} + b$ for some $a, b \in \mathbb{N}$, $\sqrt{k} > b > 0$ such that $n' \in (0, k]$. In the first case, $\alpha_*^{1D} = c\sqrt{k}/k$. In the case of 2D patching, in the worst case, $\boldsymbol{\mu}_2$ segments can be placed such that they will contribute to $c\sqrt{k}$ tokens. In this case, $\alpha_*^{2D} \geq c\sqrt{k}/k$ and $\alpha_*^{1D} \leq \alpha_*^{2D}$. If $n'$ is not a multiple of $\sqrt{k}$, the same analysis applies for the $c\sqrt{k}$ segments of $\boldsymbol{\mu}_2$. To account for the remainder $b$, we note that for any $b > 0$, in 2D case, it adds $\sqrt{k}$ label-relevant tokens to the fraction $\alpha_*^{2D}$ so that $\alpha_*^{2D} \geq \frac{c\sqrt{k}+\sqrt{k}}{k}$. In the case of 1D patching, $\alpha_*^{1D} = \frac{c\sqrt{k}+b}{k}$. Given that $b < \sqrt{k}$, this concludes the proof. $\square$

## A.3 ADDITIONAL ILLUSTRATIONS OF PROPOSITION 1

To illustrate the benefits of 2D modeling and patching, we present several examples of time series in Figure 9. We define $\boldsymbol{\mu}_1$ using functions such as log, cosine, and sine. We then set $\boldsymbol{\mu}_2 = \mathbf{1}_k$, $n' = 3$ and randomly shuffle $\boldsymbol{\mu}_1$ and $\boldsymbol{\mu}_2$ segments within the generated input time series.

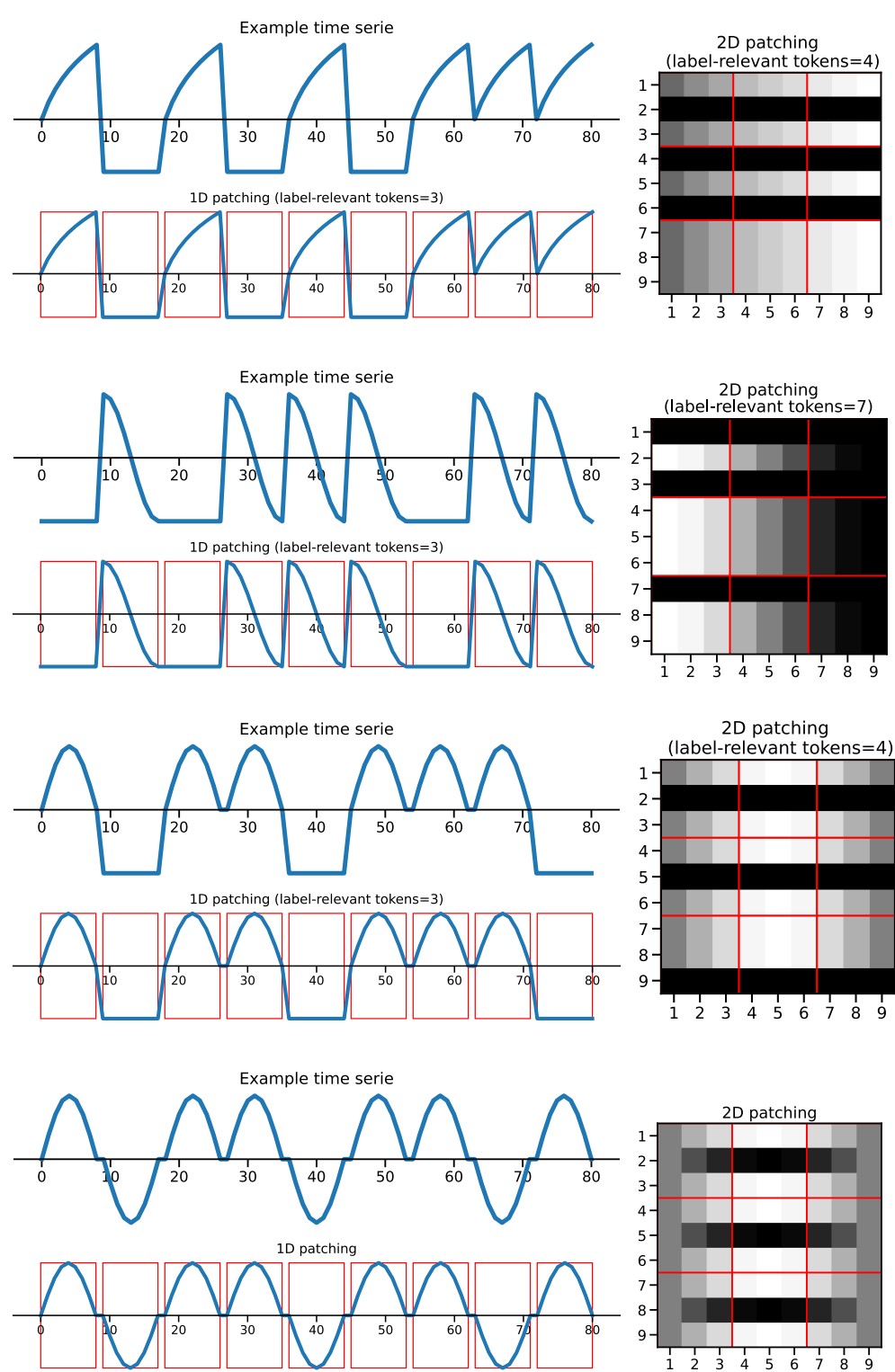

Figure 9: Illustration of Proposition 1 on more generated time series. In each example considered, 2D patching is more beneficial due the higher number of label-relevant tokens.

Table 7: Data used to pretrain Transformers for comparison of 1D and 2D patching.

| Dataset | Number of examples | Prop. of taken examples |
|---------|--------------------|-----------------------|
| ECG | 20835 | 45.7% |
| EMG | 163 | 100% |
| Epilepsy | 11480 | 100% |
| FD-A | 10912 | 100% |
| FD-B | 13619 | 100% |
| Gesture | 1320 | 100% |
| HAR | 20835 | 78.7% |
| SleepEEG | 20836 | 4.5% |

## B  DETAILS ON THE COMPARISON OF 1D AND 2D PATCHING FOR TRANSFORMERS

### B.1  ARCHITECTURE AND PRETRAINING

To evaluate the effect of 1D versus 2D patching on representations learned by Transformers, we fix the Transformer architecture and pretraining strategy, and only change the patching approach for generating input tokens. We adopt the setup of Feofanov et al. (2025) since their Transformer block implementation (ViTUnit class here) for time series classification is similar to the classical ViT. Specifically, the model comprises 6 Transformer layers, each with 8 attention heads and an embedding dimension of 256.

For pretraining, we employ contrastive learning following (Feofanov et al., 2025; He et al., 2020). The augmentation technique to generate positive pairs is RandomCropResize with a crop rate varying within $[0\%, 20\%]$. All time series are resized to a fixed length $T = 512$ using interpolation.

We examine both non-overlapping and overlapping patches following Goswami et al. (2024); Nie et al. (2023). For non-overlapping 1D patching, we generate 32 patches of size 16. For non-overlapping 2D patching, we first arrange the 1D patches in a matrix of size $32 \times 16$ and then extract 32 patches of size $2 \times 8$. After flattening, we obtain 32 patches of size 16, similar to the 1D setting, but semantically different. For overlapping 1D patching, we apply a stride of 8, which yields 64 patches of size 16. For overlapping 2D patching, we rearrange these 1D patches again in a matrix of size $64 \times 16$ and then extract 32 patches of size $4 \times 8$. Flattening yields 32 patches of size 32.

### B.2  DATASET

To pretrain the different models, we first generate a pretraining dataset from publicly available datasets that are not part of the evaluation benchmark. In detail, we consider a concatenation of the following datasets: ECG (Clifford et al., 2017), EMG (Goldberger et al., 2000), Epilepsy (Andrzejak et al., 2001), FD-A and FD-B (Lessmeier et al., 2016), Gesture (Liu et al., 2009), HAR (Anguita et al., 2013), SleepEEG (Kemp et al., 2000). To reduce computation time, we construct a subset of the full dataset containing 100 000 samples, with a sufficiently balanced distribution across the individual source datasets. We give more details in Table 7 on how many samples were taken from each dataset to form the pretraining corpus.

## C  DETAILS ON THE EXPERIMENTAL SETUP

**Datasets** UCR (Dau et al., 2019) comprises 128 univariate time series datasets of varying sample size ($16 \leq N_{\text{train}} \leq 8926$) and series length ($15 \leq T \leq 2844$). UEA (Bagnall et al., 2018) consists of 30 multivariate time series datasets. Following Feofanov et al. (2025), we exclude three datasets (AtrialFibrillation, StandWalkJump, PenDigits) from UEA in our main evaluation due to their short sequence length or small test size.

**Vision Transformers** Our study mainly examines three differently pretrained ViTs: OpenCLIP (Cherti et al., 2023; Ilharco et al., 2021), SigLIP 2 (Tschannen et al., 2025), and DINOv3 (Siméoni et al., 2025). CLIP (Radford et al., 2021) performs contrastive learning of image and text encoders on image-text pairs. We reuse the ViT image encoders of OpenCLIP (Cherti et al., 2023; Ilharco et al., 2021) models trained with the LAION-2B English subset of LAION-5B (Schuhmann et al., 2022). SigLIP 2 (Tschannen et al., 2025) adopts contrastive learning on image-text pairs, but with a Sigmoid loss, complemented by captioning-based pretraining, self-distillation, and masked prediction. In contrast, DINOv3 (Siméoni et al., 2025) is solely pretrained on images through self-distillation with a student-teacher architecture and objectives at both the image and patch level. For each pretraining approach, we consider multiple vision model sizes (ViT-B, ViT-L, ViT-H) with varying layer depth (12, 24, and 32 layers). Additionally, we investigate the effectiveness of ViTs from DINOv2 (Oquab et al., 2024) and Masked Autoencoders (He et al., 2022) in the appendix.

**Baselines** We compare TiViT to two state-of-the-art TSFMs exclusively pretrained on time series. Mantis (Feofanov et al., 2025) is a Transformer model (8 M parameters) comprising 6 layers and 8 heads per layer, pretrained on 2 million time series with contrastive learning. As stated by Feofanov et al. (2025), Mantis is based on the ViT architecture, making it particularly suitable for our comparison with large-scale ViTs trained on natural images. Moment (Goswami et al., 2024) is a family of Transformers pretrained on 13 million time series with masked modeling. In our study, we consider Moment-base with 12 layers and 125 M parameters.
We further consider GPT4TS (Zhou et al., 2023) pretrained on textual data and a wide range of supervised and self-supervised baselines (pre-)trained per time series dataset. The 9 supervised baselines comprise: ResNet (Wang et al., 2017), FCN (Wang et al., 2017), DTW (Dau et al., 2019), CNN (Zebik et al., 2017), MLP (Wang et al., 2017), Encoder (Serrà et al., 2018), TWIESN (Tanisaro & Heidemann, 2016), MCNN (Cui et al., 2016), and TimesNet (Wu et al., 2023). The 5 self-supervised baselines are: TS2Vec (Yue et al., 2022), T-Loss (Franceschi et al., 2019), TS-TCC (Eldele et al., 2021), TNC (Tonekaboni et al., 2021), and TST (Zerveas et al., 2021). For all of these baselines, we utilize the classification accuracy reported by Goswami et al. (2024) in our comparison.
Furthermore, we evaluate the effectiveness of two state-of-the-art TSFMs that have been designed for time series forecasting in time series classification: Chronos Bolt Base (Ansari et al., 2024) and VisionTS (Chen et al., 2024) with MAE Base backbone. We average the sequence of their output representations to obtain a single representation for linear classification.

**Implementation** To assess the effectiveness of TiViT and TSFM representations in time series classification, we train a logistic regressor with the LBFGS solver per dataset. Our evaluation adheres to the standard train-test splits provided by the UCR and UEA archive and reserves 20% of the train split for validation. For the time series-to-image transformation, we resize the grayscale images to the resolution expected by the ViT with nearest interpolation and adjust the contrast with a factor of $0.8$. To compute the mutual kNN alignment score between models, we select the 10 largest UCR datasets, sample 1024 time series from each dataset, and measure the overlap of their representations for k=5. This setup is in line with Huh et al. (2024). All experiments can be performed on a single NVIDIA V100 GPU with 16 GB memory.

**Anomaly detection** For this task, we equip TiViT with 6 layers of OpenCLIP ViT-B, apply no patch overlap, and flatten the sequence of representations before learning a linear reconstruction head per dataset. TiViT is evaluated across 248 dataset from the UCR Anomaly Archive (Wu & Keogh, 2023) and compared to the following baselines: Moment (Goswami et al., 2024), GPT4TS (Zhou et al., 2023), TimesNet (Wu et al., 2023), Anomaly Transformer (Xu et al., 2022), DGHL (Challu et al., 2022), and kNN (Ramaswamy et al., 2000) with $k = 5$. We utilize the adjusted best F1 score (Goswami et al., 2023; Challu et al., 2022) and VUS-ROC score (Paparrizos et al., 2022) reported for each baseline by Goswami et al. (2024).

**Forecasting** We further evaluate TiViT in long-horizon time series forecasting on 8 standard datasets (Wu et al., 2021; Ilbert et al., 2024). Similar to the best setup for anomaly detection, TiViT utilizes 6 layers of OpenCLIP ViT-B as backbone, applies no patch overlap, and flattens the sequence of representations. A linear forecasting head is learned per dataset and forecasting horizon in $\{96, 192, 336, 720\}$. Our comparison considers 8 baselines. There are 2 TSFMs evaluated with linear probing: Moment (Goswami et al., 2024) and GPT4TS (Zhou et al., 2023). Moreover, there are 6 supervised methods: PatchTST (Nie et al., 2023), DLinear (Zeng et al., 2023), TimesNet (Wu et al., 2023), FEDformer (Zhou et al., 2022), N-BEATS (Oreshkin et al., 2020), and Stationary. The

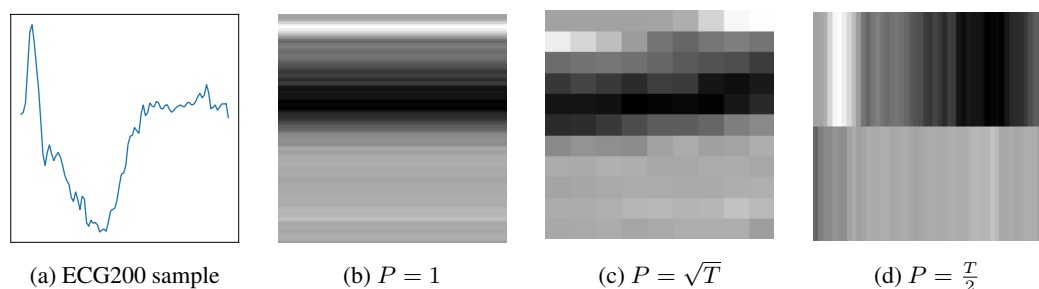

| (a) ECG200 sample | (b) $P = 1$ | (c) $P = \sqrt{T}$ | (d) $P = \frac{T}{2}$ |

Figure 10: Effect of patch size $P$ on the time series-to-image transformation of a sample from the ECG 200 (Olszewski, 2001) dataset. To match the ViT input resolution, a small patch size ($P = 1$) requires horizontal stretching, while a large patch size ($P = \frac{T}{2}$) requires vertical stretching. Both scenarios result in redundant tokens.

Table 8: Effect of patch size and overlap on validation accuracy on UCR benchmark.

| Patch size | $\sqrt{T}$ | | | $P*$ | | |
|---|---|---|---|---|---|---|
| Overlap | 0.0 | 0.5 | 0.9 | 0.0 | 0.5 | 0.9 |
| Val accuracy | 78.0 | 80.3 | 80.7 | 88.1 | 88.9 | 89.7 |
| Test accuracy | 78.3 | 80.4 | 81.6 | 79.3 | 80.7 | 81.7 |

Mean Squared Error (MSE) and Mean Absolute Error (MAE) per baseline have been reported by Goswami et al. (2024).

# D  ADDITIONAL ANALYSIS ON TIVIT

## D.1  PATCH SIZE AND OVERLAP

In Section 4.1, we report for TiViT that a patch size $P = \sqrt{T}$ and a stride $S = \frac{P}{10}$ yields high classification accuracy on any time series of length $T$. The patch size parameter $P$ affects the visual appearance of the image representation provided to the ViT for feature extraction. Figure 10 displays a time series sample from the ECG200 Olszewski (2001) dataset along with its corresponding image representations for three different patch sizes. After patching and stacking, the 2D matrix is resized to the quadratic image resolution required by ViTs. Using very small (Figure 10b) or very large (Figure 10d) patch sizes results in redundant tokens representing the same input signal. To avoid a computationally expensive hyperparameter search to find the best patch size $P*$ per dataset, we propose to select $P = \sqrt{T}$ for any dataset of length $T$. A patch size of $\sqrt{T}$ yields a square-shaped image prior to resizing and thus the most diverse set of patches without any horizontal or vertical distortion (Figure 10c). Moreover, this setting is in line with our theoretical consideration in Section 5.1.

Table 8 presents the classification accuracy for TiViT with a CLIP backbone (TiViT-CLIP) and both non-overlapping and overlapping patches. To provide an upper bound on the classification performance, we perform a hyperparameter search for the best patch size $P^*$. Specifically, for each dataset of length $T$, we consider 20 equally spaced values in $[1, \frac{T}{2}]$ and identify the patch size that maximizes classification accuracy on the validation set. Note that, while there is a small decline in accuracy in the case of no overlap, when consistently applying $P = \sqrt{T}$, the computational cost is reduced by a factor of 20. The impact of the correct patch size vanishes with increasing overlap. Figure 11 visualizes the effect of patch overlap for TiViT with CLIP, DINOv2, and SigLIP 2 backbones while fixing the patch size at $P = \sqrt{T}$. All versions of TiViT achieve high classification accuracy when utilizing an overlap of 0.9 (corresponding to stride $S = \frac{P}{10}$).

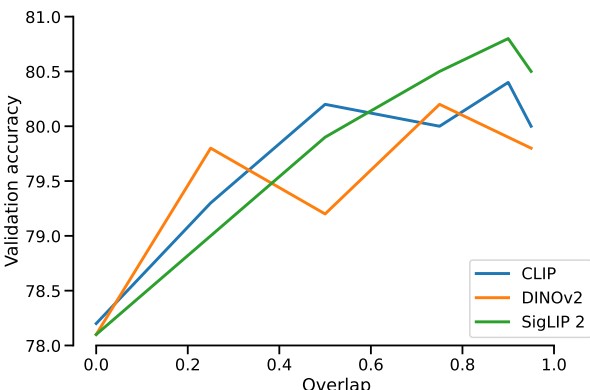

Figure 11: Effect of patch overlap on the classification accuracy of TiViT with different backbones.

Table 9: Comparison of interpolation methods on the UCR benchmark.

| Interpolation | Antialias | Accuracy |
|---|---|---|
| Bilinear | False | 81.2 |
|  | True | 80.9 |
| Bicubic | False | 79.1 |
|  | True | 79.1 |
| Lanczos | - | 80.6 |
| Nearest | - | 81.6 |

### D.2 INTERPOLATION ALGORITHM FOR IMAGE RESIZING

In our time series-to-image transformation, we resize the grayscale images to the resolution expected by the ViT with nearest interpolation by default. To further investigate the impact of the resizing method, we conduct additional experiments using bilinear and bicubic interpolation, both with and without antialiasing, and Lanczos interpolation. Table 9 summarizes our results on the UCR benchmark and indicates that nearest interpolation yields the highest classification accuracy. We hypothesize that nearest interpolation is optimal for TiViT since it preserves the raw time series signals without introducing any smoothing artifacts.

### D.3 IMAGING METHOD FOR TIME SERIES

In Section 3, we describe the transformation of time series into grayscale heatmaps, motivated by our theoretical insight in Section 5.1. Here, we explore two alternative image representations. Specifically, we visualize the time series as line plots, similar to Li et al. (2023b), and Gramian Angular Fields (GAF). We provide these 2D representations to TiViT and evaluate their effectiveness for classification on the UCR benchmark. For the two new imaging methods, we perform a hyperparameter search on the hidden layers ([10, 14, 18]) and choose the best configuration based on validation accuracy. The test accuracy is shown in Table 10. Our results indicate that TiViT achieves the highest classification accuracy using the heatmap-based representations.

### D.4 AGGREGATION OF HIDDEN TOKEN REPRESENTATIONS

As described in Section 3, we obtain a single embedding for each time series by averaging the ViT hidden representations in a particular layer. We now evaluate the performance of TiViT when using the CLS token from each layer instead. Table 11 compares the linear classification performance on the UCR dataset using either the CLS token or the mean of all tokens. To ensure a fair comparison, we determine the best performing layer for each approach based on the validation accuracy. Across

Table 10: Comparison of imaging methods on the UCR benchmark.

| Imaging method | Backbone | Layer | Accuracy |
|---|---|---|---|
| Gramian Angular Field | ViT-H/14 | 14 | 76.4 |
| Lineplot | ViT-H/14 | 14 | 80.7 |
| Heatmap | ViT-H/14 | 14 | 81.6 |

Table 11: Linear classification accuracy of TiViT on the UCR dataset with different ways of aggregating the hidden representations per layer. We report the total number of layers including the output layer and the index of the best performing layer starting from 0.

| Model | # Layers | Average of tokens | | CLS token | |
|---|---|---|---|---|---|
| | | Layer | Accuracy | Layer | Accuracy |
| TiViT-DINOv2 | 25 | 15 | 80.0 | 17 | 79.1 |
| TiViT-SigLIP 2 | 28 | 10 | 80.6 | 14 | 71.7 |
| TiViT-CLIP | 33 | 14 | **81.6** | 18 | 78.6 |

all backbones, the CLS token consistently results in lower test accuracy, confirming our choice to use the mean hidden representation in TiViT. Interestingly, the best performing CLS tokens appear in later layers compared to the best performing mean tokens. Therefore, utilizing the mean representations does not only enhance classification accuracy, but also reduce computational cost.

## D.5 INTRINSIC DIMENSION AND PRINCIPAL COMPONENTS OF HIDDEN REPRESENTATIONS

The intrinsic dimension quantifies the minimum number of variables required to represent a local neighborhood of samples in the representation space. To estimate the intrinsic dimension, the TWO-NN estimator introduced by Facco et al. (2017) leverages the distance of each data point to its first and second nearest neighbor. As noted by the authors, a larger number of data points reduces the average distance to the second neighbor, and thus increases the intrinsic dimension. To mitigate this effect, they propose to subsample the dataset. Given a dataset of size $N$, we report the intrinsic dimension for $\frac{N}{4}$ subsamples in the main paper, which is in line with Valeriani et al. (2023). In Figure 12, we compare the intrinsic dimension of average representations from hidden layers using $N$, $\frac{N}{2}$, $\frac{N}{4}$, and $\frac{N}{8}$ samples for estimation. The layer with the highest intrinsic dimension, which is central to our analysis, remains the same regardless of the subsampling ratio.

Since the intrinsic dimension only characterizes the local geometry of the representation space, we further provide a global analysis using principal components. Specifically, in Figure 13, we determine the number of principal components that are necessary to cover 95% of the variance in the data. For DINOv2, we observe a peak in the number of principal components in the middle layers that corresponds to the layers achieving the best classification accuracy. Interestingly, CLIP

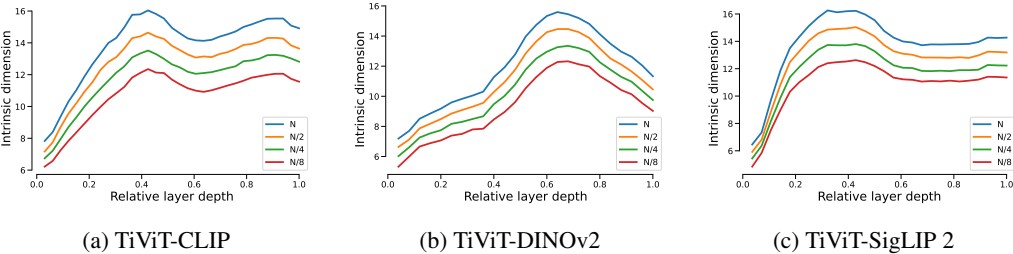

| (a) TiViT-CLIP | (b) TiViT-DINOv2 | (c) TiViT-SigLIP 2 |
|---|---|---|

Figure 12: Intrinsic dimension of hidden representations per layer from CLIP, DINOv2, and SigLIP computed for subsamples of the dataset in $\left\{N, \frac{N}{2}, \frac{N}{4}, \frac{N}{8}\right\}$.

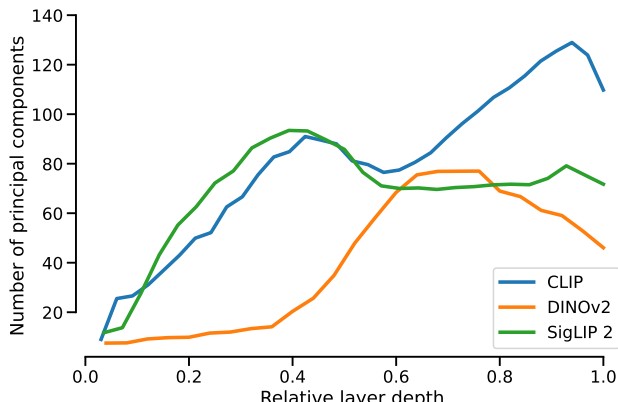

Figure 13: Number of principal components necessary to cover 95% of variance in the ViT representations per layer averaged across UCR datasets.

Table 12: Linear classification with TiViT on the UCR benchmark. For each model, we report the test accuracy achieved with the best performing hidden layer.

| Model | Architecture | Layer (Max) | Parameters | Data | Accuracy |
|-------|-------------|-------------|-----------|------|----------|
| TiViT-DINOv3 | ViT-L/14 | 17 (25) | 202 M | LVD-1689M | 80.2 |
| TiViT-SigLIP 2 | SoViT-400m/14 | 12 (28) | 184 M | WebLI (10B) | 80.6 |
| TiViT-CLIP | ViT-H/14 | 14 (33) | 257 M | LAION-2B | **81.6** |

and SigLIP 2 exhibit two peaks in the number of principal components across the layers. The middle-layers corresponding to the first peak yield the highest time series classification accuracy.

### D.6  SIZE OF VIT BACKBONE

We report the performance of TiViT with CLIP ViT-H backbone in Section 4.2 of the main paper. Table 13 provides a detailed analysis of how the performance of TiViT varies with the size of the ViT backbone, including ViT-B (with two patch sizes), ViT-L, and ViT-H. Remarkably, with only 6 Transformer layers from ViT-B, TiViT achieves an accuracy of 80.8%. While matching the number of Transformer layers in Mantis, TiViT surpasses Mantis (80.1%) in classification accuracy. However, the hidden dimensionality is higher for the ViT-B backbone used in TiViT. By utilizing a larger backbone, specifically 14 hidden layers of ViT-H/14, we achieve the highest accuracy of 81.3%, significantly outperforming conventional TSFMs.

### D.7  SIZE OF PRETRAINING DATASET

ViTs are pretrained on massive image datasets to learn rich and transferable features. These image datasets are orders of magnitude larger than the time series corpora used to pretrain models such as Mantis (2M samples) or Moment (13M samples). To investigate how the size of the ViT pretraining

Table 13: Linear classification of TiViT-CLIP with varying size of the ViT backbone. For each model, we report the test accuracy on the UCR dataset achieved with the best performing hidden layer representation and the number of parameters up to this layer.

| Architecture | Layer (total number) | Parameters | Accuracy |
|-------------|---------------------|-----------|----------|
| ViT-B/32 | 8 (13) | 52 M | 79.8 |
| ViT-B/16 | 6 (13) | 36 M | 80.8 |
| ViT-L/14 | 10 (25) | 178 M | 80.3 |
| ViT-H/14 | 14 (33) | 257 M | **81.6** |

Table 14: Comparison of CLIP-ViT-L-14 pretraining datasets on UCR benchmark.

| Dataset | Backbone | Layer | Accuracy |
|---|---|---|---|
| Laion400M | CLIP-ViT-L/14 | 10 | 81.6 |
| Laion2B | CLIP-ViT-L/14 | 10 | 80.5 |

Table 15: Comparison of different backbones and feature extraction layers on the UCR benchmark.

| Backbone | Layer | Accuracy |
|---|---|---|
| ViT-H/14 | 14 | 81.6 |
| ConvNeXt-XXLarge | 15 | 82.1 |

dataset affects the classification performance of TiViT, we compare TiViT with a CLIP-ViT-L backbone pretrained on 400M and 2B samples. As shown in Table 14, the model pretrained on 400M images outperforms the one pretrained on 2B images in time series classification. This suggests that dataset size alone does not guarantee superior performance in cross-domain tasks.

## D.8 CONVOLUTIONAL BACKBONE

We focus our study on ViTs because they are the most widely used vision backbones, trained on the largest datasets, and thus enable a comparison of different pretraining paradigms. Nonetheless, we also include a comparison with CNN-based methods. DINOv2, SigLIP 2, and MAE are exclusively built upon ViTs, and thus the only setting we can identify with a convolutional backbone (ConvNeXt) is OpenCLIP. We perform an ablation study for TiViT using different ConvNeXt layers in $\{10, 15, 20, 25\}$ and evaluate the classification accuracy on the UCR benchmark. As shown in Table 15, our method TiViT is fully compatible with pretrained convolutional models and can achieve even higher accuracies on the UCR benchmark when using a ConvNeXt backbone compared to the typical ViT.

## D.9 MASKED AUTOENCODER BACKBONE

In the main paper, we analyze the reusability of ViT backbones from CLIP Radford et al. (2021); Schuhmann et al. (2022), DINOv3 Siméoni et al. (2025), and SigLIP 2 Tschannen et al. (2025) in time series classification. In contrast, Chen et al. (2024) repurpose Masked Autoencoders (MAEs) He et al. (2022) for time series forecasting. To enable a direct comparison, we now utilize the hidden representations of MAE Base, Large, and Huge in time series classification.

Our analysis in Table 16 shows that for MAEs using the CLS token yields better performance in time series classification than averaging token representations. Moreover, Table 16 presents a comparison across MAEs of different sizes, showing that larger backbones consistently achieve higher accuracy. Different from contrastively pretrained models, summarized in Table 12 of the main paper, the best representations for time series classification with MAE lie in later layers. We further observe that the hidden representations of the later MAE layers up to the output layer perform similar in time series classification, while there is a significant gap between hidden representations and output

Table 16: Linear classification accuracy of TiViT with varying MAE backbone size and aggregation of hidden representations per layer. We report the total number of layers including the output layer and the index of the best performing layer starting from 0.

| Architecture | # Layers | Average of tokens | | CLS token | |
|---|---|---|---|---|---|
| | | Layer | Acc | Layer | Acc |
| MAE Base | 13 | 8 | 72.7 | 9 | 73.8 |
| MAE Large | 25 | 14 | 74.3 | 18 | 75.6 |
| MAE Huge | 33 | 20 | 75.9 | 20 | **76.7** |

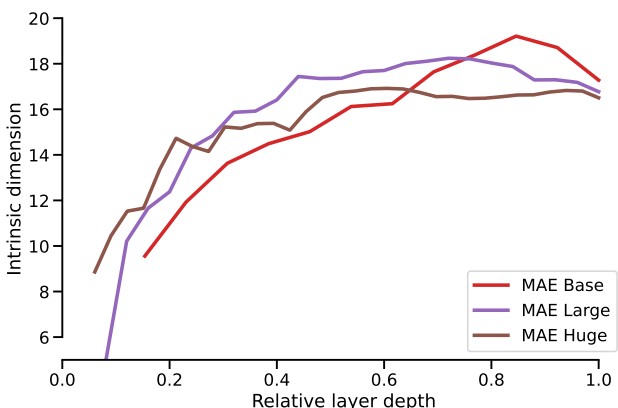

Figure 14: Intrinsic dimensionality of CLS tokens per MAE layer averaged across UCR datasets.

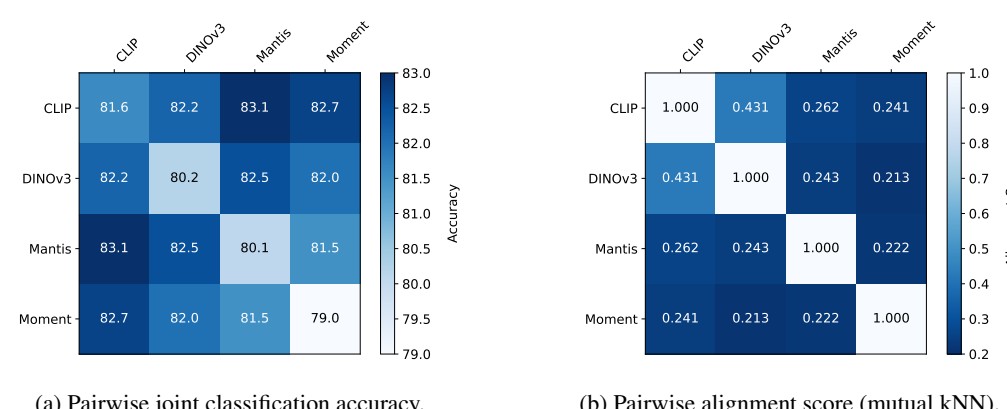

(a) Pairwise joint classification accuracy.   (b) Pairwise alignment score (mutual kNN).

Figure 15: The representations of frozen ViTs and TSFMs are concatenated and used in linear classification. Results are averaged over 128 datasets from the UCR benchmark.

representations for TiViT-CLIP (see Figure 4a in the main paper). Figure 14 illustrates the intrinsic dimension of the CLS tokens per layer averaged across the UCR datasets. We observe that the intrinsic dimension increases up to 60% of the layer depth, while the later layers mostly exhibit a similar intrinsic dimension, explaining their similar classification performance.

It is worth noting that MAE has only been pretrained on ImageNet-1k Deng et al. (2009) with 1.5 million samples, whereas CLIP has been pretrained on the significantly larger LAION-2B Schuhmann et al. (2022) dataset with 2 billion samples. We hypothesize that being exposed to a larger set of images during training enhances the capacity of a vision model to extract discriminative patterns from 2D time series representations.

### D.10    ALIGNMENT AND FUSION OF TIVIT AND TSFM REPRESENTATIONS

In Table 2 of our main paper, we report the alignment and joint classification accuracy for TiViT and TSFMs. Figure 15 is an additional visualization of the pairwise scores as heatmaps.

### D.11    FEATURE VISUALIZATION

In Section 4.4, we apply attention rollout to two samples from the ECG200 dataset, demonstrating that TiViT attends to salient regions of the time series images. Figure 17 further illustrates this behavior with three examples each from the AllGestureWiimoteX and ElectricDevices datasets, showing the original image, the corresponding attention rollout, and the overlay.

We further employ t-SNE to investigate the structure of the representations extracted by TiViT. Figure 16 presents t-SNE visualizations for 12 additional datasets. The results underscore TiViT's ability to uncover intrinsic cluster structures without access to labels and without being explicitly trained on time series.

Another way of understanding the features learned by ViTs is noise maximization. Ghiasi et al. (2022) have generated images that highly activate a particular feature in ViTs starting from random noise. TiViT applies a frozen backbone and thus utilizes the exact same features of a ViT learned from natural images. Their visualizations underline that ViT-B captures general edges and textures in early layers, and more specialized objects in later layers. Please note that TiViT only uses the first six layers of ViT-B, where there are mostly patterns and less semantic components.

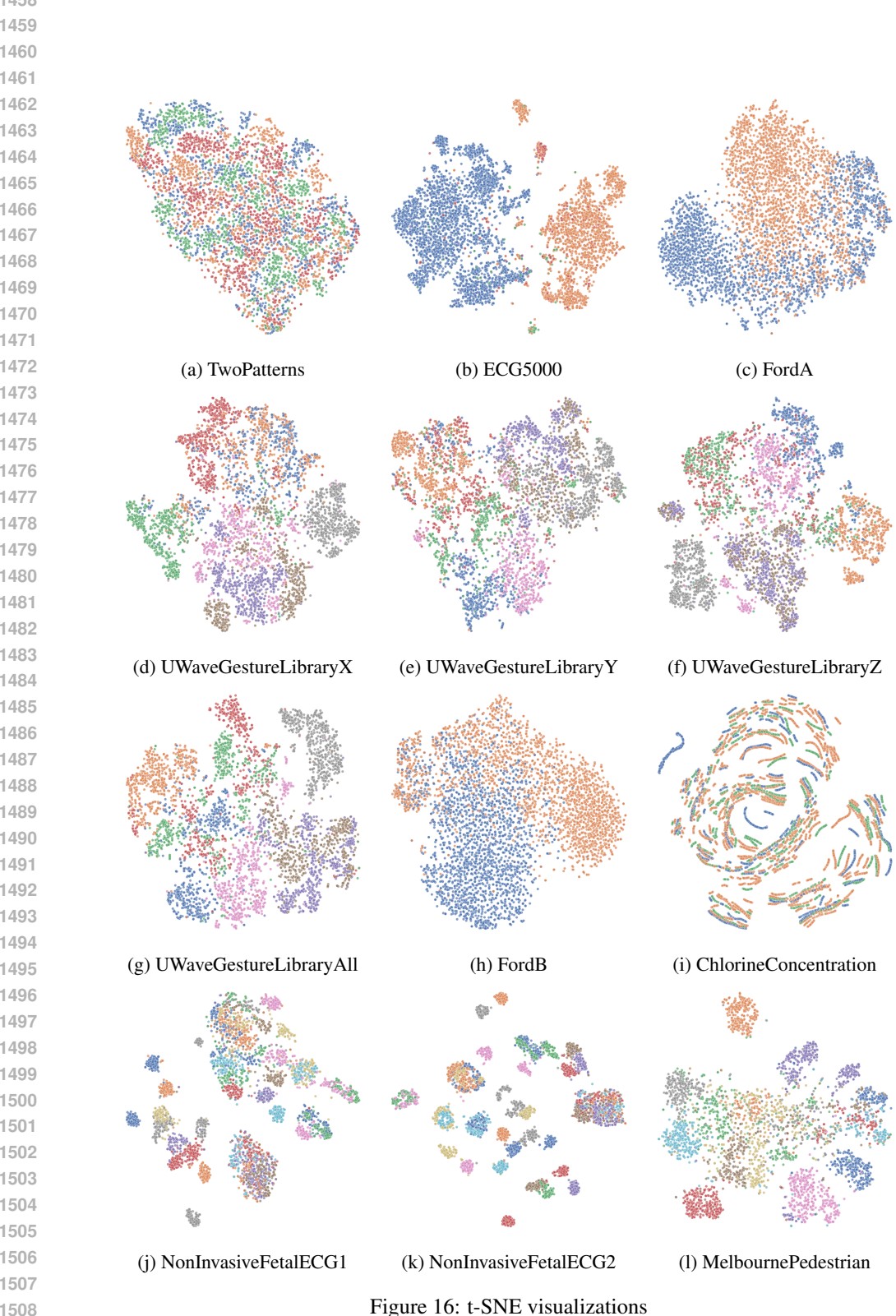

Figure 16: t-SNE visualizations

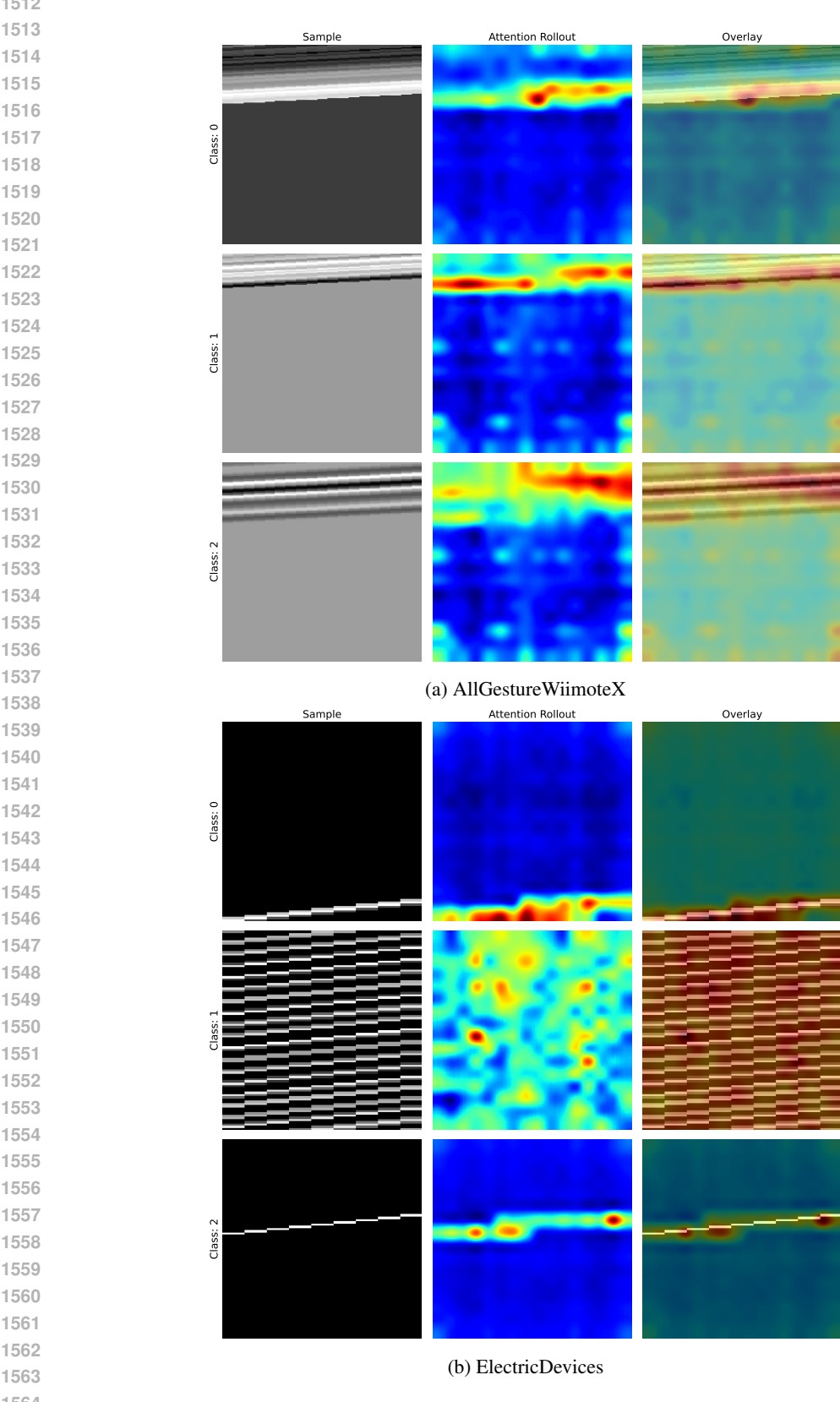

(a) AllGestureWiimoteX

(b) ElectricDevices

Figure 17: Attention rollout.

## E  DETAILED BENCHMARKING RESULTS

In the main paper, we report the average accuracy of TiViT and TSFM across 128 univariate datasets from the UCR archive and 27 multivariate datasets from the UEA archive. Here, we report the full linear classification benchmark with accuracy scores for Mantis, Moment, TiViT, and their combinations on each dataset. Table 17 presents the performance on the UCR dataset, while Table 18 reports the results on the UEA dataset. Additionally, Table 19 provides the mean rank of all five methods on both benchmarks. If multiple element share the same rank, we assign them the lowest rank in the group. Comparisons with supervised and self-supervised baselines are provided in Table 20 for the UCR benchmark and in Table 21 for the UEA benchmark.

Furthermore, we assess the performance of TiViT against baseline methods in time series forecasting on 8 standard datasets (Table 22) and in time series anomaly detection on 248 datasets from the UCR Anomaly Archive (Table 23).

Table 17: Classification accuracy for 128 univariate datasets from the UCR benchmark. We report the mean and standard deviation across three random seeds.

| Dataset | Moment | Mantis | TiViT | TiViT + Moment | TiViT + Mantis |
|---|---|---|---|---|---|
| ACSF1 | $0.673 \pm 0.012$ | $0.667 \pm 0.021$ | $\textbf{0.773} \pm 0.015$ | $\textbf{0.773} \pm 0.006$ | $0.757 \pm 0.015$ |
| Adiac | $0.728 \pm 0.004$ | $0.728 \pm 0.011$ | $0.708 \pm 0.009$ | $\textbf{0.732} \pm 0.008$ | $0.730 \pm 0.012$ |
| AllGestureWiimoteX | $0.686 \pm 0.010$ | $0.699 \pm 0.003$ | $0.685 \pm 0.010$ | $0.717 \pm 0.009$ | $\textbf{0.726} \pm 0.001$ |
| AllGestureWiimoteY | $0.710 \pm 0.006$ | $0.742 \pm 0.007$ | $0.721 \pm 0.015$ | $0.750 \pm 0.022$ | $\textbf{0.760} \pm 0.014$ |
| AllGestureWiimoteZ | $0.605 \pm 0.007$ | $0.673 \pm 0.018$ | $0.658 \pm 0.015$ | $0.690 \pm 0.014$ | $\textbf{0.700} \pm 0.014$ |
| ArrowHead | $0.804 \pm 0.012$ | $0.745 \pm 0.007$ | $0.819 \pm 0.049$ | $\textbf{0.851} \pm 0.011$ | $0.829 \pm 0.035$ |
| BME | $0.936 \pm 0.010$ | $0.991 \pm 0.010$ | $0.991 \pm 0.015$ | $0.987 \pm 0.018$ | $\textbf{0.996} \pm 0.008$ |
| Beef | $0.667 \pm 0.067$ | $0.689 \pm 0.019$ | $\textbf{0.800} \pm 0.067$ | $\textbf{0.800} \pm 0.000$ | $0.789 \pm 0.069$ |
| BeetleFly | $0.850 \pm 0.050$ | $0.867 \pm 0.058$ | $0.917 \pm 0.058$ | $0.917 \pm 0.058$ | $\textbf{0.950} \pm 0.000$ |
| BirdChicken | $0.883 \pm 0.029$ | $\textbf{0.950} \pm 0.000$ | $0.917 \pm 0.029$ | $0.900 \pm 0.000$ | $0.933 \pm 0.029$ |
| CBF | $0.907 \pm 0.030$ | $0.990 \pm 0.009$ | $\textbf{0.999} \pm 0.001$ | $0.997 \pm 0.004$ | $\textbf{0.999} \pm 0.001$ |
| Car | $0.856 \pm 0.035$ | $0.828 \pm 0.010$ | $0.844 \pm 0.010$ | $0.878 \pm 0.010$ | $\textbf{0.889} \pm 0.025$ |
| Chinatown | $0.962 \pm 0.003$ | $\textbf{0.964} \pm 0.006$ | $0.950 \pm 0.018$ | $0.954 \pm 0.025$ | $\textbf{0.964} \pm 0.010$ |
| ChlorineConcentration | $0.733 \pm 0.010$ | $0.643 \pm 0.009$ | $0.728 \pm 0.008$ | $\textbf{0.744} \pm 0.012$ | $0.738 \pm 0.000$ |
| CinCECGTorso | $0.719 \pm 0.056$ | $0.727 \pm 0.021$ | $\textbf{0.868} \pm 0.034$ | $0.837 \pm 0.063$ | $0.860 \pm 0.039$ |
| Coffee | $\textbf{1.000} \pm 0.000$ | $\textbf{1.000} \pm 0.000$ | $\textbf{1.000} \pm 0.000$ | $\textbf{1.000} \pm 0.000$ | $\textbf{1.000} \pm 0.000$ |
| Computers | $0.712 \pm 0.036$ | $0.740 \pm 0.012$ | $\textbf{0.785} \pm 0.005$ | $0.784 \pm 0.011$ | $0.781 \pm 0.023$ |
| CricketX | $0.706 \pm 0.020$ | $0.726 \pm 0.015$ | $0.753 \pm 0.006$ | $0.757 \pm 0.013$ | $\textbf{0.765} \pm 0.011$ |
| CricketY | $0.693 \pm 0.018$ | $0.732 \pm 0.017$ | $0.765 \pm 0.006$ | $0.776 \pm 0.008$ | $\textbf{0.783} \pm 0.012$ |
| CricketZ | $0.740 \pm 0.016$ | $0.721 \pm 0.009$ | $0.773 \pm 0.017$ | $0.779 \pm 0.006$ | $\textbf{0.791} \pm 0.012$ |
| Crop | $0.709 \pm 0.003$ | $0.695 \pm 0.001$ | $0.675 \pm 0.001$ | $\textbf{0.714} \pm 0.002$ | $0.707 \pm 0.002$ |
| DiatomSizeReduction | $0.900 \pm 0.030$ | $0.881 \pm 0.032$ | $\textbf{0.949} \pm 0.055$ | $0.935 \pm 0.048$ | $0.944 \pm 0.054$ |
| DistalPhalanxOutlineAgeGroup | $0.743 \pm 0.011$ | $\textbf{0.746} \pm 0.017$ | $0.703 \pm 0.015$ | $0.729 \pm 0.011$ | $0.717 \pm 0.011$ |
| DistalPhalanxOutlineCorrect | $0.762 \pm 0.017$ | $0.728 \pm 0.007$ | $\textbf{0.769} \pm 0.029$ | $0.766 \pm 0.008$ | $0.757 \pm 0.014$ |
| DistalPhalanxTW | $0.643 \pm 0.004$ | $\textbf{0.698} \pm 0.007$ | $0.640 \pm 0.012$ | $0.671 \pm 0.011$ | $0.626 \pm 0.019$ |
| DodgerLoopDay | $0.442 \pm 0.014$ | $\textbf{0.517} \pm 0.036$ | $0.488 \pm 0.043$ | $0.467 \pm 0.014$ | $0.508 \pm 0.040$ |
| DodgerLoopGame | $0.691 \pm 0.062$ | $0.720 \pm 0.018$ | $0.797 \pm 0.045$ | $0.766 \pm 0.073$ | $\textbf{0.802} \pm 0.061$ |
| DodgerLoopWeekend | $\textbf{0.986} \pm 0.013$ | $0.978 \pm 0.007$ | $0.959 \pm 0.011$ | $0.981 \pm 0.008$ | $0.969 \pm 0.015$ |
| ECG200 | $0.843 \pm 0.006$ | $0.840 \pm 0.017$ | $\textbf{0.863} \pm 0.006$ | $0.847 \pm 0.031$ | $0.847 \pm 0.021$ |
| ECG5000 | $0.934 \pm 0.002$ | $0.926 \pm 0.005$ | $0.934 \pm 0.002$ | $\textbf{0.936} \pm 0.003$ | $\textbf{0.936} \pm 0.004$ |
| ECGFiveDays | $0.919 \pm 0.059$ | $0.967 \pm 0.012$ | $0.953 \pm 0.030$ | $\textbf{0.972} \pm 0.032$ | $0.959 \pm 0.028$ |
| EOGHorizontalSignal | $0.559 \pm 0.012$ | $0.542 \pm 0.014$ | $0.598 \pm 0.008$ | $0.634 \pm 0.008$ | $\textbf{0.642} \pm 0.012$ |
| EOGVerticalSignal | $0.462 \pm 0.021$ | $\textbf{0.530} \pm 0.013$ | $0.445 \pm 0.006$ | $0.476 \pm 0.016$ | $0.471 \pm 0.008$ |
| Earthquakes | $\textbf{0.734} \pm 0.025$ | $0.707 \pm 0.018$ | $0.698 \pm 0.007$ | $0.717 \pm 0.008$ | $0.703 \pm 0.017$ |
| ElectricDevices | $0.626 \pm 0.006$ | $0.698 \pm 0.003$ | $\textbf{0.757} \pm 0.009$ | $0.741 \pm 0.003$ | $0.748 \pm 0.007$ |
| EthanolLevel | $\textbf{0.649} \pm 0.008$ | $0.433 \pm 0.004$ | $0.574 \pm 0.008$ | $0.617 \pm 0.013$ | $0.586 \pm 0.008$ |
| FaceAll | $0.724 \pm 0.006$ | $\textbf{0.797} \pm 0.007$ | $0.741 \pm 0.005$ | $0.743 \pm 0.005$ | $0.762 \pm 0.007$ |
| FaceFour | $0.826 \pm 0.076$ | $\textbf{0.958} \pm 0.007$ | $0.871 \pm 0.029$ | $0.909 \pm 0.034$ | $0.936 \pm 0.035$ |
| FacesUCR | $0.789 \pm 0.010$ | $0.888 \pm 0.003$ | $0.881 \pm 0.007$ | $0.881 \pm 0.004$ | $\textbf{0.912} \pm 0.004$ |
| FiftyWords | $0.733 \pm 0.015$ | $0.736 \pm 0.010$ | $0.758 \pm 0.013$ | $0.788 \pm 0.003$ | $\textbf{0.796} \pm 0.006$ |
| Fish | $0.949 \pm 0.000$ | $0.954 \pm 0.000$ | $0.952 \pm 0.007$ | $0.945 \pm 0.020$ | $\textbf{0.968} \pm 0.013$ |
| FordA | $0.915 \pm 0.002$ | $0.910 \pm 0.003$ | $0.915 \pm 0.003$ | $\textbf{0.927} \pm 0.004$ | $0.917 \pm 0.000$ |
| FordB | $0.801 \pm 0.004$ | $0.769 \pm 0.002$ | $\textbf{0.812} \pm 0.005$ | $0.809 \pm 0.007$ | $0.800 \pm 0.012$ |
| FreezerRegularTrain | $0.973 \pm 0.011$ | $0.976 \pm 0.012$ | $\textbf{0.997} \pm 0.002$ | $0.996 \pm 0.005$ | $\textbf{0.997} \pm 0.002$ |
| FreezerSmallTrain | $0.840 \pm 0.012$ | $0.870 \pm 0.020$ | $\textbf{0.992} \pm 0.004$ | $0.982 \pm 0.006$ | $0.990 \pm 0.003$ |
| Fungi | $0.753 \pm 0.033$ | $0.810 \pm 0.025$ | $0.787 \pm 0.022$ | $0.806 \pm 0.014$ | $\textbf{0.812} \pm 0.023$ |
| GestureMidAirD1 | $0.659 \pm 0.012$ | $0.664 \pm 0.027$ | $0.746 \pm 0.013$ | $0.731 \pm 0.023$ | $\textbf{0.756} \pm 0.032$ |
| GestureMidAirD2 | $0.567 \pm 0.016$ | $0.585 \pm 0.040$ | $0.667 \pm 0.012$ | $0.644 \pm 0.032$ | $\textbf{0.669} \pm 0.015$ |
| GestureMidAirD3 | $0.359 \pm 0.019$ | $0.392 \pm 0.013$ | $\textbf{0.472} \pm 0.016$ | $0.449 \pm 0.016$ | $0.464 \pm 0.025$ |
| GesturePebbleZ1 | $0.893 \pm 0.015$ | $0.917 \pm 0.003$ | $0.895 \pm 0.006$ | $0.924 \pm 0.000$ | $\textbf{0.928} \pm 0.003$ |
| GesturePebbleZ2 | $0.846 \pm 0.018$ | $\textbf{0.895} \pm 0.007$ | $0.840 \pm 0.010$ | $0.861 \pm 0.035$ | $0.892 \pm 0.017$ |
| GunPoint | $0.984 \pm 0.027$ | $0.987 \pm 0.007$ | $\textbf{0.996} \pm 0.004$ | $0.987 \pm 0.012$ | $\textbf{0.996} \pm 0.004$ |
| GunPointAgeSpan | $0.980 \pm 0.002$ | $\textbf{0.998} \pm 0.002$ | $0.992 \pm 0.002$ | $0.993 \pm 0.002$ | $0.994 \pm 0.008$ |
| GunPointMaleVersusFemale | $\textbf{1.000} \pm 0.000$ | $0.999 \pm 0.002$ | $0.996 \pm 0.002$ | $\textbf{1.000} \pm 0.000$ | $\textbf{1.000} \pm 0.000$ |
| GunPointOldVersusYoung | $\textbf{1.000} \pm 0.000$ | $\textbf{1.000} \pm 0.000$ | $0.988 \pm 0.002$ | $\textbf{1.000} \pm 0.000$ | $\textbf{1.000} \pm 0.000$ |
| Ham | $\textbf{0.752} \pm 0.025$ | $0.667 \pm 0.010$ | $0.695 \pm 0.040$ | $0.721 \pm 0.024$ | $0.724 \pm 0.019$ |
| HandOutlines | $0.930 \pm 0.007$ | $0.931 \pm 0.006$ | $0.936 \pm 0.007$ | $\textbf{0.945} \pm 0.010$ | $0.932 \pm 0.007$ |
| Haptics | $0.491 \pm 0.026$ | $0.462 \pm 0.002$ | $0.498 \pm 0.007$ | $0.535 \pm 0.040$ | $\textbf{0.539} \pm 0.009$ |
| Herring | $\textbf{0.698} \pm 0.018$ | $0.682 \pm 0.024$ | $0.599 \pm 0.009$ | $0.630 \pm 0.039$ | $0.625 \pm 0.027$ |
| HouseTwenty | $0.947 \pm 0.010$ | $0.961 \pm 0.010$ | $0.972 \pm 0.005$ | $0.972 \pm 0.010$ | $\textbf{0.980} \pm 0.005$ |
| InlineSkate | $0.364 \pm 0.019$ | $0.334 \pm 0.021$ | $0.398 \pm 0.015$ | $0.401 \pm 0.006$ | $\textbf{0.408} \pm 0.015$ |
| InsectEPGRegularTrain | $0.987 \pm 0.014$ | $\textbf{1.000} \pm 0.000$ | $\textbf{1.000} \pm 0.000$ | $\textbf{1.000} \pm 0.000$ | $\textbf{1.000} \pm 0.000$ |

| Dataset | Moment | Mantis | TiViT | TiViT + Moment | TiViT + Mantis |
|---|---|---|---|---|---|
| | | Continuation of Table 17 | | | |
| InsectEPGSmallTrain | $0.953 \pm 0.008$ | $\mathbf{1.000} \pm 0.000$ | $0.968 \pm 0.007$ | $0.973 \pm 0.005$ | $0.999 \pm 0.002$ |
| InsectWingbeatSound | $0.539 \pm 0.003$ | $0.470 \pm 0.019$ | $0.536 \pm 0.015$ | $\mathbf{0.560} \pm 0.007$ | $0.539 \pm 0.010$ |
| ItalyPowerDemand | $\mathbf{0.938} \pm 0.005$ | $0.910 \pm 0.006$ | $0.920 \pm 0.018$ | $0.936 \pm 0.011$ | $0.923 \pm 0.018$ |
| LargeKitchenAppliances | $0.859 \pm 0.005$ | $0.820 \pm 0.010$ | $\mathbf{0.883} \pm 0.014$ | $0.873 \pm 0.018$ | $0.879 \pm 0.014$ |
| Lightning2 | $0.760 \pm 0.041$ | $0.781 \pm 0.025$ | $0.803 \pm 0.028$ | $\mathbf{0.820} \pm 0.028$ | $0.803 \pm 0.016$ |
| Lightning7 | $0.836 \pm 0.036$ | $0.749 \pm 0.021$ | $0.831 \pm 0.021$ | $\mathbf{0.881} \pm 0.008$ | $0.822 \pm 0.024$ |
| Mallat | $0.915 \pm 0.010$ | $0.868 \pm 0.028$ | $0.956 \pm 0.017$ | $\mathbf{0.963} \pm 0.016$ | $0.958 \pm 0.018$ |
| Meat | $0.911 \pm 0.038$ | $\mathbf{0.939} \pm 0.019$ | $0.800 \pm 0.000$ | $0.900 \pm 0.029$ | $0.850 \pm 0.044$ |
| MedicalImages | $0.730 \pm 0.003$ | $0.707 \pm 0.024$ | $0.740 \pm 0.006$ | $\mathbf{0.780} \pm 0.006$ | $0.761 \pm 0.014$ |
| MelbournePedestrian | $\mathbf{0.933} \pm 0.003$ | $0.908 \pm 0.005$ | $0.862 \pm 0.006$ | $0.932 \pm 0.005$ | $0.925 \pm 0.003$ |
| MiddlePhalanxOutlineAgeGroup | $0.489 \pm 0.029$ | $\mathbf{0.587} \pm 0.019$ | $0.537 \pm 0.036$ | $0.530 \pm 0.004$ | $0.571 \pm 0.023$ |
| MiddlePhalanxOutlineCorrect | $0.816 \pm 0.009$ | $\mathbf{0.845} \pm 0.009$ | $0.789 \pm 0.015$ | $0.792 \pm 0.016$ | $0.805 \pm 0.016$ |
| MiddlePhalanxTW | $0.506 \pm 0.019$ | $0.442 \pm 0.017$ | $0.506 \pm 0.023$ | $0.498 \pm 0.025$ | $\mathbf{0.511} \pm 0.010$ |
| MixedShapesRegularTrain | $0.947 \pm 0.004$ | $0.955 \pm 0.006$ | $0.974 \pm 0.002$ | $0.973 \pm 0.003$ | $\mathbf{0.976} \pm 0.002$ |
| MixedShapesSmallTrain | $0.882 \pm 0.004$ | $0.904 \pm 0.002$ | $0.950 \pm 0.002$ | $0.937 \pm 0.004$ | $\mathbf{0.957} \pm 0.003$ |
| MoteStrain | $0.889 \pm 0.028$ | $0.895 \pm 0.026$ | $0.875 \pm 0.021$ | $\mathbf{0.918} \pm 0.008$ | $0.901 \pm 0.025$ |
| NonInvasiveFetalECGThorax1 | $0.919 \pm 0.002$ | $0.797 \pm 0.006$ | $0.884 \pm 0.004$ | $\mathbf{0.924} \pm 0.003$ | $0.885 \pm 0.009$ |
| NonInvasiveFetalECGThorax2 | $0.927 \pm 0.002$ | $0.817 \pm 0.004$ | $0.915 \pm 0.001$ | $\mathbf{0.934} \pm 0.004$ | $0.918 \pm 0.005$ |
| OSULeaf | $0.917 \pm 0.004$ | $0.899 \pm 0.005$ | $0.977 \pm 0.006$ | $0.972 \pm 0.010$ | $\mathbf{0.978} \pm 0.009$ |
| OliveOil | $\mathbf{0.856} \pm 0.051$ | $0.822 \pm 0.107$ | $0.656 \pm 0.077$ | $0.778 \pm 0.019$ | $0.711 \pm 0.051$ |
| PLAID | $0.775 \pm 0.017$ | $0.852 \pm 0.001$ | $0.888 \pm 0.008$ | $0.901 \pm 0.011$ | $\mathbf{0.928} \pm 0.012$ |
| PhalangesOutlinesCorrect | $\mathbf{0.795} \pm 0.006$ | $0.794 \pm 0.008$ | $0.789 \pm 0.004$ | $\mathbf{0.795} \pm 0.008$ | $0.787 \pm 0.004$ |
| Phoneme | $0.277 \pm 0.003$ | $0.293 \pm 0.008$ | $0.377 \pm 0.006$ | $0.372 \pm 0.003$ | $\mathbf{0.386} \pm 0.006$ |
| PickupGestureWiimoteZ | $0.713 \pm 0.042$ | $0.767 \pm 0.023$ | $0.887 \pm 0.031$ | $0.847 \pm 0.046$ | $\mathbf{0.893} \pm 0.023$ |
| PigAirwayPressure | $0.109 \pm 0.007$ | $0.588 \pm 0.012$ | $0.540 \pm 0.006$ | $0.447 \pm 0.013$ | $\mathbf{0.598} \pm 0.010$ |
| PigArtPressure | $0.780 \pm 0.010$ | $0.827 \pm 0.017$ | $0.817 \pm 0.013$ | $0.833 \pm 0.019$ | $\mathbf{0.846} \pm 0.005$ |
| PigCVP | $0.747 \pm 0.027$ | $0.753 \pm 0.007$ | $0.702 \pm 0.019$ | $0.761 \pm 0.018$ | $\mathbf{0.801} \pm 0.012$ |
| Plane | $0.997 \pm 0.005$ | $\mathbf{1.000} \pm 0.000$ | $\mathbf{1.000} \pm 0.000$ | $\mathbf{1.000} \pm 0.000$ | $\mathbf{1.000} \pm 0.000$ |
| PowerCons | $0.931 \pm 0.006$ | $0.933 \pm 0.010$ | $0.894 \pm 0.022$ | $\mathbf{0.943} \pm 0.013$ | $0.906 \pm 0.020$ |
| ProximalPhalanxOutlineAgeGroup | $0.802 \pm 0.020$ | $\mathbf{0.852} \pm 0.007$ | $0.833 \pm 0.027$ | $0.824 \pm 0.005$ | $0.828 \pm 0.017$ |
| ProximalPhalanxOutlineCorrect | $0.883 \pm 0.010$ | $\mathbf{0.885} \pm 0.008$ | $0.861 \pm 0.020$ | $0.871 \pm 0.016$ | $0.858 \pm 0.023$ |
| ProximalPhalanxTW | $\mathbf{0.767} \pm 0.010$ | $0.740 \pm 0.015$ | $0.751 \pm 0.022$ | $0.730 \pm 0.010$ | $0.759 \pm 0.023$ |
| RefrigerationDevices | $0.496 \pm 0.017$ | $0.526 \pm 0.022$ | $0.555 \pm 0.007$ | $0.531 \pm 0.005$ | $\mathbf{0.570} \pm 0.014$ |
| Rock | $0.727 \pm 0.031$ | $0.700 \pm 0.060$ | $\mathbf{0.873} \pm 0.099$ | $\mathbf{0.873} \pm 0.115$ | $0.853 \pm 0.117$ |
| ScreenType | $0.499 \pm 0.020$ | $0.468 \pm 0.026$ | $0.530 \pm 0.014$ | $0.516 \pm 0.002$ | $\mathbf{0.552} \pm 0.027$ |
| SemgHandGenderCh2 | $0.761 \pm 0.018$ | $0.883 \pm 0.006$ | $0.879 \pm 0.001$ | $0.878 \pm 0.013$ | $\mathbf{0.914} \pm 0.006$ |
| SemgHandMovementCh2 | $0.398 \pm 0.010$ | $0.654 \pm 0.018$ | $0.545 \pm 0.016$ | $0.538 \pm 0.031$ | $\mathbf{0.688} \pm 0.024$ |
| SemgHandSubjectCh2 | $0.648 \pm 0.013$ | $0.826 \pm 0.005$ | $0.840 \pm 0.002$ | $0.838 \pm 0.012$ | $\mathbf{0.895} \pm 0.007$ |
| ShakeGestureWiimoteZ | $0.887 \pm 0.012$ | $0.867 \pm 0.012$ | $0.827 \pm 0.031$ | $\mathbf{0.907} \pm 0.031$ | $0.840 \pm 0.020$ |
| ShapeletSim | $0.967 \pm 0.010$ | $0.919 \pm 0.012$ | $\mathbf{1.000} \pm 0.000$ | $\mathbf{1.000} \pm 0.000$ | $\mathbf{1.000} \pm 0.000$ |
| ShapesAll | $0.886 \pm 0.003$ | $0.844 \pm 0.010$ | $0.901 \pm 0.003$ | $\mathbf{0.913} \pm 0.008$ | $0.908 \pm 0.007$ |
| SmallKitchenAppliances | $0.733 \pm 0.010$ | $0.796 \pm 0.013$ | $\mathbf{0.830} \pm 0.003$ | $0.817 \pm 0.018$ | $0.812 \pm 0.008$ |
| SmoothSubspace | $0.898 \pm 0.023$ | $\mathbf{0.971} \pm 0.004$ | $0.956 \pm 0.010$ | $0.964 \pm 0.010$ | $\mathbf{0.971} \pm 0.010$ |
| SonyAIBORobotSurface1 | $0.834 \pm 0.013$ | $0.858 \pm 0.015$ | $0.890 \pm 0.012$ | $0.869 \pm 0.009$ | $\mathbf{0.896} \pm 0.010$ |
| SonyAIBORobotSurface2 | $0.855 \pm 0.027$ | $0.895 \pm 0.012$ | $0.911 \pm 0.049$ | $0.914 \pm 0.049$ | $\mathbf{0.923} \pm 0.048$ |
| StarLightCurves | $0.969 \pm 0.003$ | $0.968 \pm 0.002$ | $0.973 \pm 0.002$ | $\mathbf{0.976} \pm 0.002$ | $\mathbf{0.976} \pm 0.002$ |
| Strawberry | $\mathbf{0.972} \pm 0.002$ | $0.960 \pm 0.004$ | $0.959 \pm 0.002$ | $0.968 \pm 0.006$ | $0.959 \pm 0.003$ |
| SwedishLeaf | $0.915 \pm 0.007$ | $0.942 \pm 0.006$ | $0.955 \pm 0.003$ | $\mathbf{0.959} \pm 0.006$ | $0.958 \pm 0.003$ |
| Symbols | $0.957 \pm 0.019$ | $0.957 \pm 0.031$ | $0.966 \pm 0.034$ | $\mathbf{0.973} \pm 0.020$ | $0.967 \pm 0.035$ |
| SyntheticControl | $0.966 \pm 0.004$ | $0.992 \pm 0.002$ | $0.999 \pm 0.002$ | $0.993 \pm 0.003$ | $\mathbf{1.000} \pm 0.000$ |
| ToeSegmentation1 | $\mathbf{0.963} \pm 0.007$ | $0.952 \pm 0.012$ | $0.952 \pm 0.012$ | $\mathbf{0.963} \pm 0.005$ | $0.959 \pm 0.009$ |
| ToeSegmentation2 | $0.885 \pm 0.015$ | $\mathbf{0.954} \pm 0.008$ | $0.923 \pm 0.008$ | $0.895 \pm 0.027$ | $0.926 \pm 0.004$ |
| Trace | $\mathbf{1.000} \pm 0.000$ | $\mathbf{1.000} \pm 0.000$ | $\mathbf{1.000} \pm 0.000$ | $\mathbf{1.000} \pm 0.000$ | $\mathbf{1.000} \pm 0.000$ |
| TwoLeadECG | $0.901 \pm 0.020$ | $0.998 \pm 0.002$ | $0.997 \pm 0.001$ | $0.997 \pm 0.001$ | $\mathbf{1.000} \pm 0.000$ |
| TwoPatterns | $0.989 \pm 0.001$ | $0.946 \pm 0.007$ | $0.998 \pm 0.000$ | $\mathbf{0.999} \pm 0.001$ | $0.998 \pm 0.001$ |
| UMD | $\mathbf{0.993} \pm 0.000$ | $\mathbf{0.993} \pm 0.000$ | $\mathbf{0.993} \pm 0.000$ | $\mathbf{0.993} \pm 0.000$ | $\mathbf{0.993} \pm 0.000$ |
| UWaveGestureLibraryAll | $0.923 \pm 0.002$ | $0.874 \pm 0.004$ | $0.940 \pm 0.001$ | $\mathbf{0.950} \pm 0.005$ | $0.944 \pm 0.003$ |
| UWaveGestureLibraryX | $0.792 \pm 0.001$ | $0.779 \pm 0.004$ | $0.828 \pm 0.004$ | $\mathbf{0.838} \pm 0.004$ | $\mathbf{0.838} \pm 0.002$ |
| UWaveGestureLibraryY | $0.711 \pm 0.006$ | $0.678 \pm 0.009$ | $0.749 \pm 0.004$ | $0.758 \pm 0.004$ | $\mathbf{0.763} \pm 0.006$ |
| UWaveGestureLibraryZ | $0.731 \pm 0.001$ | $0.742 \pm 0.009$ | $0.770 \pm 0.003$ | $0.772 \pm 0.004$ | $\mathbf{0.786} \pm 0.001$ |
| Wafer | $0.992 \pm 0.002$ | $0.996 \pm 0.000$ | $\mathbf{1.000} \pm 0.000$ | $\mathbf{1.000} \pm 0.000$ | $\mathbf{1.000} \pm 0.000$ |
| Wine | $\mathbf{0.889} \pm 0.019$ | $0.796 \pm 0.037$ | $0.599 \pm 0.065$ | $0.747 \pm 0.028$ | $0.759 \pm 0.049$ |
| WordSynonyms | $0.655 \pm 0.003$ | $0.626 \pm 0.017$ | $0.649 \pm 0.007$ | $\mathbf{0.690} \pm 0.005$ | $0.681 \pm 0.006$ |
| Worms | $0.745 \pm 0.033$ | $0.710 \pm 0.033$ | $0.762 \pm 0.027$ | $\mathbf{0.805} \pm 0.026$ | $0.762 \pm 0.052$ |
| WormsTwoClass | $0.775 \pm 0.037$ | $0.745 \pm 0.007$ | $0.784 \pm 0.020$ | $\mathbf{0.792} \pm 0.026$ | $0.766 \pm 0.022$ |
| Yoga | $0.833 \pm 0.008$ | $0.771 \pm 0.014$ | $0.826 \pm 0.009$ | $\mathbf{0.852} \pm 0.007$ | $0.844 \pm 0.007$ |
| | | End of Table | | | |

Table 18: Classification accuracy for 27 multivariate datasets from the UEA benchmark. We report the mean and standard deviation across three random seeds.

| Dataset | Moment | Mantis | TiViT | TiViT + Moment | TiViT + Mantis |
|---|---|---|---|---|---|
| ArticularyWordRecognition | 0.988 ± 0.002 | **0.991** ± 0.002 | 0.977 ± 0.003 | 0.977 ± 0.003 | 0.974 ± 0.005 |
| BasicMotions | **1.000** ± 0.000 | **1.000** ± 0.000 | **1.000** ± 0.000 | **1.000** ± 0.000 | **1.000** ± 0.000 |
| CharacterTrajectories | **0.982** ± 0.001 | 0.973 ± 0.001 | 0.964 ± 0.005 | **0.982** ± 0.001 | 0.978 ± 0.005 |
| Cricket | **1.000** ± 0.000 | 0.986 ± 0.000 | **1.000** ± 0.000 | **1.000** ± 0.000 | **1.000** ± 0.000 |
| DuckDuckGeese | **0.467** ± 0.081 | 0.433 ± 0.023 | 0.393 ± 0.081 | 0.413 ± 0.064 | 0.433 ± 0.050 |
| ERing | 0.895 ± 0.022 | 0.905 ± 0.025 | 0.975 ± 0.014 | 0.977 ± 0.006 | **0.981** ± 0.007 |
| EigenWorms | 0.746 ± 0.022 | 0.746 ± 0.016 | **0.911** ± 0.016 | 0.880 ± 0.009 | **0.911** ± 0.012 |
| Epilepsy | **1.000** ± 0.000 | 0.990 ± 0.004 | **1.000** ± 0.000 | **1.000** ± 0.000 | **1.000** ± 0.000 |
| EthanolConcentration | 0.445 ± 0.013 | 0.269 ± 0.044 | **0.485** ± 0.012 | 0.473 ± 0.030 | 0.465 ± 0.019 |
| FaceDetection | 0.584 ± 0.007 | 0.592 ± 0.006 | 0.598 ± 0.004 | 0.584 ± 0.007 | **0.607** ± 0.005 |
| FingerMovements | **0.633** ± 0.045 | 0.593 ± 0.025 | 0.517 ± 0.040 | 0.620 ± 0.036 | 0.553 ± 0.050 |
| HandMovementDirection | **0.279** ± 0.051 | 0.212 ± 0.021 | 0.275 ± 0.016 | 0.257 ± 0.036 | 0.257 ± 0.027 |
| Handwriting | 0.296 ± 0.018 | **0.425** ± 0.013 | 0.307 ± 0.034 | 0.340 ± 0.002 | 0.385 ± 0.021 |
| Heartbeat | 0.735 ± 0.007 | **0.800** ± 0.017 | 0.732 ± 0.008 | 0.717 ± 0.022 | 0.769 ± 0.003 |
| InsectWingbeat | 0.231 ± 0.012 | **0.573** ± 0.012 | 0.355 ± 0.008 | 0.332 ± 0.018 | 0.443 ± 0.020 |
| JapaneseVowels | 0.918 ± 0.006 | **0.978** ± 0.003 | 0.940 ± 0.002 | 0.938 ± 0.012 | 0.933 ± 0.008 |
| LSST | 0.571 ± 0.005 | 0.607 ± 0.009 | 0.604 ± 0.005 | 0.610 ± 0.009 | **0.652** ± 0.003 |
| Libras | 0.861 ± 0.017 | 0.887 ± 0.026 | 0.907 ± 0.006 | **0.922** ± 0.022 | 0.920 ± 0.018 |
| MotorImagery | 0.530 ± 0.026 | **0.563** ± 0.012 | **0.563** ± 0.049 | 0.560 ± 0.044 | 0.553 ± 0.042 |
| NATOPS | 0.900 ± 0.029 | **0.931** ± 0.014 | 0.869 ± 0.006 | 0.889 ± 0.006 | 0.878 ± 0.006 |
| PEMS-SF | 0.705 ± 0.029 | **0.788** ± 0.029 | 0.709 ± 0.084 | 0.763 ± 0.044 | 0.742 ± 0.087 |
| PhonemeSpectra | 0.186 ± 0.004 | 0.272 ± 0.006 | 0.245 ± 0.007 | 0.265 ± 0.007 | **0.286** ± 0.008 |
| RacketSports | 0.829 ± 0.007 | **0.919** ± 0.004 | 0.846 ± 0.010 | 0.871 ± 0.008 | 0.879 ± 0.027 |
| SelfRegulationSCP1 | 0.762 ± 0.010 | 0.825 ± 0.022 | 0.858 ± 0.008 | 0.840 ± 0.003 | **0.891** ± 0.010 |
| SelfRegulationSCP2 | 0.509 ± 0.031 | 0.491 ± 0.018 | **0.526** ± 0.038 | 0.506 ± 0.017 | 0.517 ± 0.020 |
| SpokenArabicDigits | **0.981** ± 0.003 | 0.907 ± 0.006 | 0.969 ± 0.001 | 0.979 ± 0.003 | 0.972 ± 0.002 |
| UWaveGestureLibrary | 0.846 ± 0.010 | 0.879 ± 0.015 | 0.910 ± 0.005 | 0.902 ± 0.004 | **0.919** ± 0.009 |

Table 19: Mean rank of TiViT and TSFMs across datasets from the UCR and UEA archive.

| Model | UCR | UEA |
|---|---|---|
| Moment | 3.75 | 3.33 |
| Mantis | 3.43 | 2.85 |
| TiViT *(Ours)* | 2.97 | 2.85 |
| TiViT + Moment *(Ours)* | 2.20 | 2.63 |
| TiViT + Mantis *(Ours)* | **1.95** | **2.22** |

# F BROADER IMPACTS

Since this paper presents foundational machine learning research, we do not see any direct societal risks. The broader impact of our work will depend on its specific application.

We demonstrate that our method TiViT significantly improves classification accuracy. This advancement can be beneficial in healthcare where the analysis of physiological signals is crucial for early diagnosis and treatment or in industry where the accurate monitoring of sensor data enables predictive maintenance and reduces downtime.

However, deep learning models including TiViT operate as black boxes with limited interpretability. In safety-critical domains or applications directly impacting humans, such models necessitate careful deployment and oversight. Further research into interpretability and human-in-the-loop frameworks is essential to make deep learning models trustworthy for real-world settings.

Table 20: Classification accuracy across 91 UCR datasets. Baselines from Goswami et al. (2024).

| Accuracy | TiViT + Mantis | TiViT | Mantis | MOMENT | TimesNet | GPT4TS | TS2Vec | T-Loss | TNC | TS-TCC |
|---|---|---|---|---|---|---|---|---|---|---|
| Mean | 0.848 | 0.834 | 0.826 | 0.794 | 0.572 | 0.566 | **0.851** | 0.833 | 0.786 | 0.793 |
| Median | **0.880** | 0.849 | 0.852 | 0.815 | 0.565 | 0.583 | 0.871 | 0.849 | 0.788 | 0.802 |
| Std. | 0.133 | 0.136 | 0.143 | 0.147 | 0.238 | 0.234 | 0.134 | 0.136 | 0.168 | 0.176 |

| Accuracy | TST | CNN | Encoder | FCN | MCNN | MLP | ResNet | t-LeNet | TWIESN | DTW |
|---|---|---|---|---|---|---|---|---|---|---|
| Mean | 0.658 | 0.751 | 0.743 | 0.809 | 0.702 | 0.750 | 0.825 | 0.348 | 0.726 | 0.764 |
| Median | 0.720 | 0.773 | 0.753 | 0.837 | 0.718 | 0.766 | 0.852 | 0.333 | 0.724 | 0.768 |
| Std. | 0.220 | 0.180 | 0.159 | 0.188 | 0.194 | 0.169 | 0.177 | 0.221 | 0.164 | 0.152 |

Table 21: Classification accuracy across 29 UEA datasets. Baselines from Goswami et al. (2024).

| Accuracy | TiViT + Mantis | TiViT | Mantis | MOMENT | TS2Vec | T-Loss | TNC | TS-TCC | TST | DTW |
|---|---|---|---|---|---|---|---|---|---|---|
| Mean | **71.9** | 70.6 | 69.3 | 0.670 | 0.694 | 0.646 | 0.660 | 0.657 | 0.605 | 0.638 |
| Median | **82.3** | 78.9 | 78.8 | 0.722 | 0.683 | 0.676 | 0.746 | 0.751 | 0.620 | 0.664 |
| Std. | 26.0 | 26.6 | 26.6 | 0.274 | 0.255 | 0.296 | 0.267 | 0.263 | 0.294 | 0.296 |

| Method | | Pretraining + linear probing | | | | | Supervised training | | | | | | | | | | |
|---|---|---|---|---|---|---|---|---|---|---|---|---|---|---|---|---|---|
| | | TiViT (Ours) | | MOMENT | | GPT4TS | | PatchTST | | DLinear | | TimesNet | | FEDFormer | | Stationary | | N-BEATS |
| Metric | | MSE | MAE | MSE | MAE | MSE | MAE | MSE | MAE | MSE | MAE | MSE | MAE | MSE | MAE | MSE | MAE | MSE | MAE |
| Weather | 96 | 0.153 | 0.211 | 0.154 | 0.209 | 0.162 | 0.212 | 0.149 | 0.198 | 0.176 | 0.237 | 0.172 | 0.220 | 0.217 | 0.296 | 0.173 | 0.223 | 0.152 | 0.210 |
| | 192 | 0.196 | 0.247 | 0.197 | 0.248 | 0.204 | 0.248 | 0.194 | 0.241 | 0.220 | 0.282 | 0.219 | 0.261 | 0.276 | 0.336 | 0.245 | 0.285 | 0.199 | 0.260 |
| | 336 | 0.248 | 0.285 | 0.246 | 0.285 | 0.254 | 0.286 | 0.245 | 0.282 | 0.265 | 0.319 | 0.280 | 0.306 | 0.339 | 0.380 | 0.321 | 0.338 | 0.258 | 0.311 |
| | 720 | 0.321 | 0.337 | 0.315 | 0.336 | 0.326 | 0.337 | 0.314 | 0.334 | 0.333 | 0.362 | 0.365 | 0.359 | 0.403 | 0.428 | 0.414 | 0.410 | 0.331 | 0.359 |
| ECL | 96 | 0.140 | 0.240 | 0.136 | 0.233 | 0.139 | 0.238 | 0.129 | 0.222 | 0.140 | 0.237 | 0.168 | 0.272 | 0.193 | 0.308 | 0.169 | 0.273 | 0.131 | 0.228 |
| | 192 | 0.152 | 0.251 | 0.152 | 0.247 | 0.153 | 0.251 | 0.157 | 0.240 | 0.153 | 0.249 | 0.184 | 0.289 | 0.201 | 0.315 | 0.182 | 0.286 | 0.153 | 0.248 |
| | 336 | 0.168 | 0.267 | 0.167 | 0.264 | 0.169 | 0.266 | 0.163 | 0.259 | 0.169 | 0.267 | 0.198 | 0.300 | 0.214 | 0.329 | 0.200 | 0.304 | 0.170 | 0.267 |
| | 720 | 0.204 | 0.297 | 0.205 | 0.295 | 0.206 | 0.297 | 0.197 | 0.290 | 0.203 | 0.301 | 0.220 | 0.320 | 0.246 | 0.355 | 0.222 | 0.321 | 0.208 | 0.298 |
| Traffic | 96 | 0.384 | 0.274 | 0.391 | 0.282 | 0.388 | 0.282 | 0.360 | 0.249 | 0.410 | 0.282 | 0.593 | 0.321 | 0.587 | 0.366 | 0.612 | 0.338 | 0.375 | 0.259 |
| | 192 | 0.398 | 0.280 | 0.404 | 0.287 | 0.407 | 0.290 | 0.379 | 0.256 | 0.423 | 0.287 | 0.617 | 0.336 | 0.604 | 0.373 | 0.613 | 0.340 | 0.403 | 0.274 |
| | 336 | 0.407 | 0.285 | 0.414 | 0.292 | 0.412 | 0.294 | 0.392 | 0.264 | 0.436 | 0.296 | 0.629 | 0.336 | 0.621 | 0.383 | 0.618 | 0.328 | 0.426 | 0.285 |
| | 720 | 0.443 | 0.303 | 0.450 | 0.310 | 0.450 | 0.312 | 0.432 | 0.286 | 0.466 | 0.315 | 0.640 | 0.350 | 0.626 | 0.382 | 0.653 | 0.355 | 0.508 | 0.335 |
| ETTh1 | 96 | 0.391 | 0.417 | 0.387 | 0.410 | 0.376 | 0.397 | 0.370 | 0.399 | 0.375 | 0.399 | 0.384 | 0.402 | 0.376 | 0.419 | 0.513 | 0.491 | 0.399 | 0.428 |
| | 192 | 0.411 | 0.430 | 0.410 | 0.426 | 0.416 | 0.418 | 0.413 | 0.421 | 0.405 | 0.416 | 0.436 | 0.429 | 0.420 | 0.448 | 0.534 | 0.504 | 0.451 | 0.464 |
| | 336 | 0.425 | 0.442 | 0.422 | 0.437 | 0.442 | 0.433 | 0.422 | 0.436 | 0.439 | 0.443 | 0.491 | 0.469 | 0.459 | 0.465 | 0.588 | 0.535 | 0.498 | 0.500 |
| | 720 | 0.447 | 0.469 | 0.454 | 0.472 | 0.477 | 0.456 | 0.447 | 0.466 | 0.472 | 0.490 | 0.521 | 0.500 | 0.506 | 0.507 | 0.643 | 0.616 | 0.608 | 0.573 |
| ETTh2 | 96 | 0.319 | 0.375 | 0.288 | 0.345 | 0.285 | 0.342 | 0.274 | 0.336 | 0.289 | 0.353 | 0.340 | 0.374 | 0.358 | 0.397 | 0.476 | 0.458 | 0.327 | 0.387 |
| | 192 | 0.363 | 0.406 | 0.349 | 0.386 | 0.354 | 0.389 | 0.339 | 0.379 | 0.383 | 0.418 | 0.402 | 0.414 | 0.429 | 0.439 | 0.512 | 0.493 | 0.400 | 0.435 |
| | 336 | 0.372 | 0.418 | 0.369 | 0.408 | 0.373 | 0.407 | 0.329 | 0.380 | 0.448 | 0.465 | 0.452 | 0.452 | 0.496 | 0.487 | 0.552 | 0.551 | 0.747 | 0.599 |
| | 720 | 0.407 | 0.447 | 0.403 | 0.439 | 0.406 | 0.441 | 0.379 | 0.422 | 0.605 | 0.551 | 0.462 | 0.468 | 0.463 | 0.474 | 0.562 | 0.560 | 1.454 | 0.847 |
| ETTm1 | 96 | 0.315 | 0.367 | 0.293 | 0.349 | 0.292 | 0.346 | 0.290 | 0.342 | 0.299 | 0.343 | 0.338 | 0.375 | 0.379 | 0.419 | 0.386 | 0.398 | 0.318 | 0.367 |
| | 192 | 0.352 | 0.387 | 0.326 | 0.368 | 0.332 | 0.372 | 0.332 | 0.369 | 0.335 | 0.365 | 0.374 | 0.387 | 0.426 | 0.441 | 0.459 | 0.444 | 0.355 | 0.391 |
| | 336 | 0.381 | 0.404 | 0.352 | 0.384 | 0.366 | 0.394 | 0.366 | 0.392 | 0.369 | 0.386 | 0.410 | 0.411 | 0.445 | 0.459 | 0.495 | 0.464 | 0.401 | 0.419 |
| | 720 | 0.437 | 0.436 | 0.405 | 0.416 | 0.417 | 0.421 | 0.416 | 0.420 | 0.425 | 0.421 | 0.478 | 0.450 | 0.543 | 0.490 | 0.585 | 0.516 | 0.448 | 0.448 |
| ETTm2 | 96 | 0.189 | 0.277 | 0.170 | 0.260 | 0.173 | 0.262 | 0.165 | 0.255 | 0.167 | 0.269 | 0.187 | 0.267 | 0.203 | 0.287 | 0.192 | 0.274 | 0.197 | 0.271 |
| | 192 | 0.252 | 0.318 | 0.227 | 0.297 | 0.229 | 0.301 | 0.220 | 0.292 | 0.224 | 0.303 | 0.249 | 0.309 | 0.269 | 0.328 | 0.280 | 0.339 | 0.285 | 0.328 |
| | 336 | 0.301 | 0.351 | 0.275 | 0.328 | 0.286 | 0.341 | 0.274 | 0.329 | 0.281 | 0.342 | 0.321 | 0.351 | 0.325 | 0.366 | 0.334 | 0.361 | 0.338 | 0.366 |
| | 720 | 0.382 | 0.402 | 0.363 | 0.387 | 0.378 | 0.401 | 0.362 | 0.385 | 0.397 | 0.421 | 0.408 | 0.403 | 0.421 | 0.415 | 0.417 | 0.413 | 0.395 | 0.419 |
| ILI | 24 | 2.822 | 1.142 | 2.728 | 1.114 | 2.063 | 0.881 | 1.319 | 0.754 | 2.215 | 1.081 | 2.317 | 0.934 | 3.228 | 1.260 | 2.294 | 0.945 | 4.539 | 1.528 |
| | 36 | 2.862 | 1.143 | 2.669 | 1.092 | 1.868 | 0.892 | 1.430 | 0.834 | 1.963 | 0.963 | 1.972 | 0.920 | 2.679 | 1.080 | 1.825 | 0.848 | 4.628 | 1.534 |
| | 48 | 2.846 | 1.123 | 2.728 | 1.098 | 1.790 | 0.884 | 1.553 | 0.815 | 2.130 | 1.024 | 2.238 | 0.940 | 2.622 | 1.078 | 2.010 | 0.900 | 4.957 | 1.585 |
| | 60 | 3.023 | 1.155 | 2.883 | 1.126 | 1.979 | 0.957 | 1.470 | 0.788 | 2.368 | 1.096 | 2.027 | 0.928 | 2.857 | 1.157 | 2.178 | 0.963 | 5.429 | 1.661 |

Table 22: Long-term forecasting. Baselines from Goswami et al. (2024).

| | Adjusted Best $F_1$ | | | | | VUS-ROC | | | | |
|---|---|---|---|---|---|---|---|---|---|---|
| | TiViT | Anomaly TF | Moment | GPT4TS | TimesNet | TiViT | AnomalyTF | Moment | GPT4TS | TimesNet |
| 1sddb40 | **0.935** | 0.030 | 0.540 | 0.190 | 0.680 | **0.772** | 0.640 | 0.750 | 0.660 | 0.720 |
| BIDMC1 | **1.000** | 0.990 | **1.000** | **1.000** | **1.000** | 0.642 | 0.690 | 0.650 | 0.630 | **0.740** |
| CHARISfive | 0.046 | 0.010 | **0.130** | 0.020 | 0.080 | **0.572** | 0.360 | 0.400 | 0.450 | 0.460 |
| CHARISten | **0.851** | 0.020 | 0.110 | 0.100 | 0.030 | **0.597** | 0.430 | 0.540 | 0.510 | 0.530 |
| CIMIS44AirTemperature3 | **1.000** | 0.060 | 0.980 | 0.180 | 0.470 | **0.843** | 0.640 | 0.750 | 0.620 | 0.740 |
| CIMIS44AirTemperature5 | **1.000** | 0.390 | 0.990 | 0.200 | 0.710 | **0.859** | 0.780 | 0.810 | 0.560 | 0.720 |
| ECG2 | **1.000** | **1.000** | **1.000** | 0.900 | **1.000** | 0.821 | 0.830 | **0.840** | 0.780 | 0.600 |
| ECG3 | **1.000** | 0.360 | 0.980 | 0.840 | 0.480 | **0.808** | 0.540 | 0.770 | 0.450 | 0.610 |
| Fantasia | **1.000** | 0.750 | 0.950 | 0.870 | 0.550 | **0.786** | 0.730 | 0.640 | 0.650 | 0.610 |
| GP711MarkerLFM5z4 | **1.000** | 0.930 | **1.000** | 0.640 | 0.950 | **0.886** | 0.540 | 0.730 | 0.620 | 0.720 |
| GP711MarkerLFM5z5 | **1.000** | 0.760 | 0.970 | 0.480 | 0.900 | **0.961** | 0.690 | 0.720 | 0.630 | 0.840 |
| InternalBleeding5 | **1.000** | 0.940 | **1.000** | 0.920 | **1.000** | 0.932 | 0.460 | 0.690 | 0.630 | **0.940** |
| Italianpowerdemand | 0.310 | 0.010 | **0.740** | 0.010 | 0.440 | 0.709 | 0.450 | **0.770** | 0.480 | 0.710 |
| Lab2Cmac011215EPG5 | **1.000** | 0.990 | 0.980 | 0.600 | 0.990 | 0.739 | **0.770** | 0.630 | 0.640 | 0.610 |
| Lab2Cmac011215EPG6 | 0.267 | **0.410** | 0.100 | 0.100 | 0.170 | 0.554 | **0.700** | 0.480 | 0.520 | 0.450 |
| MesoplodonDensirostris | **1.000** | **1.000** | 0.840 | **1.000** | **1.000** | 0.748 | **0.850** | 0.720 | 0.690 | 0.790 |
| PowerDemand1 | **0.994** | 0.870 | 0.440 | 0.760 | 0.950 | **0.919** | 0.720 | 0.540 | 0.600 | 0.750 |
| TkeepFirstMARS | **0.577** | 0.010 | 0.150 | 0.020 | 0.230 | 0.728 | 0.520 | 0.760 | 0.500 | **0.790** |
| TkeepSecondMARS | **1.000** | 0.830 | **1.000** | 0.120 | 0.950 | **0.989** | 0.720 | 0.910 | 0.810 | 0.980 |
| WalkingAceleration5 | 0.967 | 0.990 | **1.000** | 0.870 | 0.930 | **0.968** | 0.940 | 0.870 | 0.910 | 0.850 |
| apneaecg | **0.814** | 0.400 | 0.200 | 0.310 | 0.260 | 0.608 | 0.580 | 0.690 | 0.580 | **0.760** |
| apneaecg2 | **1.000** | 0.650 | **1.000** | **1.000** | 0.650 | **0.845** | 0.790 | 0.740 | 0.650 | 0.610 |
| gait1 | **1.000** | 0.180 | 0.360 | 0.410 | 0.520 | **0.887** | 0.630 | 0.570 | 0.580 | 0.600 |
| gaitHunt1 | **0.596** | 0.080 | 0.430 | 0.100 | 0.300 | **0.847** | 0.810 | 0.680 | 0.710 | 0.840 |
| insectEPG2 | **0.962** | 0.120 | 0.230 | 0.810 | 0.960 | **0.871** | 0.650 | 0.820 | 0.560 | 0.730 |
| insectEPG4 | 0.513 | 0.980 | **1.000** | 0.210 | 0.850 | 0.691 | 0.690 | **0.720** | 0.490 | 0.650 |
| ltstdbs30791AS | **1.000** | **1.000** | **1.000** | **1.000** | **1.000** | **0.959** | 0.780 | 0.810 | 0.740 | 0.670 |
| mit14046longtermecg | 0.676 | 0.450 | 0.590 | 0.580 | 0.600 | 0.661 | 0.790 | 0.660 | 0.610 | **0.840** |
| park3m | **1.000** | 0.150 | 0.640 | 0.630 | 0.930 | **0.875** | 0.630 | 0.780 | 0.540 | 0.780 |
| qtdbSel1005V | **0.844** | 0.410 | 0.650 | 0.390 | 0.530 | 0.612 | 0.520 | **0.640** | 0.610 | 0.540 |
| qtdbSel100MLII | **1.000** | 0.420 | 0.840 | 0.600 | 0.870 | 0.573 | 0.620 | 0.620 | 0.580 | **0.650** |
| resperation1 | **0.308** | 0.000 | 0.150 | 0.010 | 0.030 | 0.725 | **0.750** | 0.670 | 0.470 | 0.670 |
| s20101mML2 | **1.000** | 0.690 | 0.710 | 0.050 | 0.080 | **0.942** | 0.640 | 0.720 | 0.640 | 0.690 |
| sddb49 | **1.000** | 0.890 | **1.000** | 0.940 | **1.000** | **0.937** | 0.660 | 0.730 | 0.580 | 0.680 |
| sel840mECG1 | **0.984** | 0.160 | 0.660 | 0.210 | 0.360 | 0.702 | 0.620 | **0.720** | 0.650 | 0.600 |
| sel840mECG2 | **0.984** | 0.150 | 0.390 | 0.280 | 0.210 | 0.683 | 0.590 | **0.690** | 0.520 | 0.520 |
| tilt12744mtable | 0.254 | 0.070 | 0.240 | 0.000 | 0.030 | 0.761 | 0.480 | 0.740 | 0.510 | 0.640 |
| tilt12754table | 0.131 | 0.230 | **0.640** | 0.060 | 0.050 | **0.855** | 0.600 | 0.820 | 0.550 | 0.750 |
| tiltAPB2 | **1.000** | 0.920 | 0.980 | 0.830 | 0.380 | **0.844** | 0.770 | 0.770 | 0.600 | 0.700 |
| tiltAPB3 | 0.148 | 0.170 | **0.850** | 0.050 | 0.090 | 0.769 | 0.680 | 0.650 | 0.440 | 0.580 |
| weallwalk | **0.706** | 0.000 | 0.580 | 0.130 | 0.170 | 0.849 | 0.730 | **0.930** | 0.870 | 0.850 |
| Mean | **0.802** | 0.475 | 0.684 | 0.449 | 0.570 | **0.789** | 0.659 | 0.711 | 0.605 | 0.695 |
| Median | **0.994** | 0.410 | 0.740 | 0.390 | 0.550 | **0.808** | 0.660 | 0.720 | 0.600 | 0.700 |
| Std | 0.300 | 0.379 | 0.321 | 0.358 | 0.355 | 0.122 | 0.124 | 0.106 | 0.107 | 0.118 |

Table 23: Anomaly detection performance across 41 datasets from the UCR Anomaly Archive measured using adjusted best $F_1$ and VUS-ROC. Bold indicates the best performance per dataset/metric. Baselines from Goswami et al. (2024).

