# OpenReview forum: "TiViT: Time Series Representations Lie Hidden in Pretrained Vision Transformers"
_ICLR.cc/2026/Conference — Submitted to ICLR 2026_

### Official Review · Reviewer_Rurf · 2025-10-24

**Soundness:** 2
**Presentation:** 2
**Contribution:** 2
**Rating:** 2
**Confidence:** 4

**Summary:**

This paper proposes TiViT (Time Vision Transformer), a framework that converts time series into two-dimensional images to leverage pretrained Vision Transformers (ViTs) for time series analysis. The authors show that frozen ViTs can outperform existing time-series foundation models on classification and anomaly detection benchmarks without any retraining or fine-tuning on time series data.

**Strengths:**

1. The paper explores transferring pretrained vision models to time series tasks, which is conceptually interesting and inspiring.
2. The experiments are comprehensive, covering multiple popular time series datasets and different pretrained ViT models.

**Weaknesses:**

1. The idea of using ViTs for time series modeling has already been widely explored. The proposed method of converting series into images via heatmaps, applying ViTs for classification, and freezing ViT parameters has been investigated before. The paper mainly reuses this framework with different pretrained backbones, lacking substantial methodological novelty.
2. As a “foundation model”, the paper only evaluates classification and anomaly detection, while omitting the more fundamental time series forecasting tasks, making the evaluation scope incomplete.
3. Pretrained ViTs are typically computationally expensive, yet the paper does not discuss time or computational complexity.
4. The time-series-to-image transformation does not introduce new information; it simply adapts the data to ViT’s input format by channel replication, but may result in redundant computation due to copy channel and interpolation-based upsampling.

**Questions:**

1. The authors claim that the model requires no training, yet it still includes a trainable linear classifier. Is the performance improvement mainly due to the pretrained ViT representations or the linear layer? Does the model truly support zero-shot inference?
2. Why do the authors choose $\sqrt{T}$ as the patch size? How does this differ from using periodicity-based patching strategies? Can the authors provide comparative results for different patch sizes?
3. The transformation from time series to images seems to add no semantic information and is mainly for input alignment with ViTs. Could a lighter transformation be used to reduce parameters and computational overhead?
4. During the image transformation, the authors replicate the same single channel three times instead of using multivariate time-series variables as separate channels. Would this design waste potential cross-variable information?

---

> ### Author Response · Authors · 2025-11-17
>
> We are pleased that the reviewer recognizes our “comprehensive" evaluation and finds our work “interesting and inspiring”. The reviewer’s concern about the novelty of our work (W1) is addressed in our general comment.
>
> > W2: “As a “foundation model”, the paper only evaluates classification and anomaly detection, while omitting the more fundamental time series forecasting tasks.”
>
> We would like to ask the reviewer why they consider time series forecasting “more fundamental” than classification or anomaly detection?
>
> VisionTS and Chronos are TSFMs for time series forecasting. Similarly, TiViT serves as a foundation model for time series classification. Owing to the large-scale contrastive pre-training of its ViT backbone, TiViT is particularly well suited for discriminative tasks.
>
> Nevertheless, we examine the effectiveness of TiViT in time series forecasting on 8 datasets in Section 4.6. As displayed in Table 22, TiViT reaches linear probing performance comparable to Moment, particularly on the Weather, Electricity, and Traffic datasets.
>
> > W3: “Pretrained ViTs are typically computationally expensive, yet the paper does not discuss time or computational complexity.”
>
> We present the number of model parameters in Table 12. TiViT with an OpenCLIP ViT-H backbone contains 257 M parameters. In Appendix D.6, we study the size of the ViT backbone and show that with only 6 Transformer layers of ViT-B (36 M parameters), TiViT achieves an accuracy of 80.8%, outperforming both TSFMs Mantis and Moment. The number of parameters of our baselines Mantis (8 M) and Moment (125 M) is mentioned in the experimental setup in Appendix C.
>
> > W4: “The time-series-to-image transformation does not introduce new information; [...] may result in redundant computation due to copy channel and interpolation-based upsampling.”
>
> We refer the reviewer to our study on the patching strategy in Section 4.1 and Appendix D.1. As displayed in Figure 10, a very small or large patch size would indeed create redundant visual tokens during resizing to the ViT input resolution. However, our patch size of sqrt(T) yields a square-shaped image prior to resizing and thus the most diverse set of patches without any distortion.
>
> > Q1: “The authors claim that the model requires no training, yet it still includes a trainable linear classifier. Is the performance improvement mainly due to the pretrained ViT representations or the linear layer? Does the model truly support zero-shot inference?”
>
> Our model does not require pre-training or end-to-end fine-tuning on time series. For the evaluation of TiViT in time series classification, we stick to the standard protocol with a linear classifier (Feofanov et al., 2025; Goswami et al., 2024).
> Since we ensure that all models (TiViT, Mantis, Moment) are evaluated with the same linear classifier, the performance improvements of TiViT can be attributed to the ViT representations.
>
> Moreover, in Section 4.5, we perform classification with a nearest centroid classifier that does not involve any learnable parameters. As displayed in Table 3, TiViT reaches the same accuracy as Mantis (77.7%) and their joint classification yields SOTA performance (80.1%).
>
> > Q2: “Why do the authors choose sqrt(T) as the patch size? How does this differ from using periodicity-based patching strategies? Can the authors provide comparative results for different patch sizes?”
>
> The reviewer has expressed their concern about redundant computation due to interpolation-based upsampling in W4. Our choice of sqrt(T) as patch size addresses this concern (see Section 4.1, Appendix D.1).
>
> Furthermore, we perform a hyperparameter search for the best patch size P* and patch overlap. Figure 3 and Table 8 show that such an exhaustive search for the optimal patch size does not yield any significant performance benefits on the UCR test set, but would only increase the computational cost.
>
> > Q3: “The transformation from time series to images seems to add no semantic information and is mainly for input alignment with ViTs. Could a lighter transformation be used to reduce parameters and computational overhead?”
>
> Our time series-to-image transformation is very lightweight. As detailed in Section 3, the transformation does not involve any learnable parameters. It only stacks 1D values into a 2D matrix, which is then resized and interpolated to the ViT input resolution.
>
> > Q4: “During the image transformation, the authors replicate the same single channel three times instead of using multivariate time-series variables as separate channels. Would this design waste potential cross-variable information?”
>
> Our approach of modeling time series channels independently is in line with existing TSFMs (Mantis, Moment). Encoding multivariate time series variables as separate image channels would only be feasible for datasets with at most three variables. This would restrict the applicability of our method since many multivariate time series contain more than three channels.

---

> > ### Comment · Reviewer_Rurf · 2025-11-17
> >
> > Thank you for the authors’ response.
> >
> > W1: In the Official Comment, the authors mentioned the “simplicity of an idea” when addressing the novelty concerns. I agree that simplicity can sometimes lead to innovation; however, the core idea of this work is not only simple but also not new. As far as I know, the survey “Harnessing Vision Models for Time Series Analysis”$^{[1]}$ already includes approaches that use ViT for classification. Moreover, it is difficult to claim that this paper “provides the first theoretical justification.”
> >
> > W2: The authors refer to comparisons in Table 22. Based on my reading, the proposed model still underperforms even compared to older baselines such as “PatchTST”. Could the authors comment on the reasons behind this?
> >
> > W3: Thank you for the additional information. However, I would also like to ask about the computational overhead in terms of runtime. If possible, could the authors provide a quantitative analysis?
> >
> > W4: The authors may have misunderstood my previous question. To my knowledge, ViT typically takes 224×224 images as input, whereas the authors’ method uses $\sqrt{T} < 224$ and therefore requires upsampling to 224, which introduces repeated pixels.
> >
> > Q1: Does the linear classifier mentioned by the authors require training? Where do its parameters come from?
> >
> > Q4: Have the authors considered fusing channels before forming the final 3-channel representation, for example by using a convolution layer to aggregate information across channels?
> >
> > [1] Ni et al., "Harnessing Vision Models for Time Series Analysis: A Survey", IJCAI 2025.

---

> > > ### Author Response · Authors · 2025-11-19
> > >
> > > We thank the reviewer for their follow-up comments and appreciate the opportunity to further clarify central aspects of our paper and our previous responses.
> > >
> > > > W1: the core idea of this work is not only simple but also not new. As far as I know, the survey “Harnessing Vision Models for Time Series Analysis” already includes approaches that use ViT for classification. Moreover, it is difficult to claim that this paper “provides the first theoretical justification.”
> > >
> > > Characterizing our work solely as an application of ViTs to time series classification does not capture its scope. We kindly refer the reviewer to our general comment on novelty and invite them to consider the following points when evaluating the contributions of our work:
> > > - How can our approach outperform any of the existing vision-based approaches (+10.1% over VisionTS, +26.3% over TimesNet) and even the most recent TSFMs (+1.5% over Mantis on the full UCR benchmark) by a significant margin without any new ideas? Which prior work has achieved such results?
> > > - Which prior work evaluates the effectiveness of large-scale vision (DINOv2/v3) and vision-language models (CLIP, SigLIPv2) for time series classification and anomaly detection? Is there any mention of a VLM being state-of-the-art in classification in the survey mentioned by the reviewer?
> > > - Which prior work finds that the hidden representations of frozen vision and vision language models are the most effective for time series classification, and reveals a correlation with their intrinsic dimension?
> > > - Which prior work analyzes the complementarity of representations extracted by ViTs and TSFMs and demonstrates consistent measurable gains in joint classification that future work can leverage to boost the performance of TSFMs?
> > > - Which prior work offers a theoretical explanation at the patch level for the benefits of 2D modeling over 1D modeling in time series classification?
> > > - Which prior work compares pretraining of Transformers on time series with 1D vs 2D modeling?
> > >
> > > None of these contributions appear in the cited survey paper, nor in any prior work that we are aware of, which demonstrates the novelty of our work.
> > >
> > > > W2: The authors refer to comparisons in Table 22. Based on my reading, the proposed model still underperforms even compared to older baselines such as “PatchTST”. Could the authors comment on the reasons behind this?
> > >
> > > As we stated in our paper and our previous response, VisionTS and Chronos are TSFMs for **forecasting** (with weak performance in classification, see Figure 5), whereas TiViT is a TSFM for **classification**. The ViT backbone of TiViT has been pretrained contrastively and is therefore well-suited for discriminative tasks such as classification. Conversely, as shown in the VisionTS paper, pretraining with masked modeling may better fit generative tasks such as time series forecasting.
> > >
> > > Forecasting is not the focus of our work and is only included for completeness, as noted in Section 4 (l. 172). We do not claim SOTA performance on this task. Nevertheless, TiViT reaches the performance of other TSFMs such as Moment and GPT4TS on several datasets of the forecasting benchmark. As mentioned in our experimental setup (l.1072), TSFMs (TiViT, Moment, GPT4TS) are evaluated with linear probing, while the other approaches, including PatchTST in Table 22, are fully supervised and trained end-to-end per dataset. This may explain performance gaps.

---

> > > > ### Author Response · Authors · 2025-11-19
> > > >
> > > > > W3: I would also like to ask about the computational overhead in terms of runtime. If possible, could the authors provide a quantitative analysis?
> > > >
> > > > We thank the reviewer for the suggestion to include a quantitative runtime analysis. We measure the time required by different methods to compute representations for all UCR datasets, comprising a total of 191,158 time series samples. To ensure a fair comparison, we conduct all measurements on the same hardware (Nvidia H100 80 GB GPU) with a fixed batch size of 256. Results are summarized below:
> > > >
> > > > |Method|Total time (seconds) |Time/sample (milliseconds)|
> > > > |-|:-:|:-:|
> > > > |TiViT ViT-B|128.90|0.67|
> > > > |TiViT ViT-H|985.19|5.15|
> > > > |VisionTS|164.19|0.86|
> > > > |Moment|105.08|0.55|
> > > > |Mantis|49.75|0.26|
> > > >
> > > > TiViT requires more time than the TSFMs Mantis and Moment for computing time series representations. However, it is noteworthy that TiViT with a CLIP-ViT-B backbone is close to the runtime of Moment while achieving significantly better performance on the UCR benchmark: 80.8% for TiViT with ViT-B (see Table 13) vs. 79.0 % for Moment (see Table 1). Furthermore, TiViT with ViT-B is not only significantly better than VisionTS (see Figure 5) but also faster. Even for our largest and strongest model (TiViT with CLIP-ViT-H backbone) the runtime differences per sample are in the order of milliseconds. We believe that for any application that is not real-time, e.g. medical diagnosis of ECG, these runtime differences are negligible and the actual classification accuracy would become the primary concern. Here, we refer again to our state-of-the-art performance reported in Section 4.2.
> > > >
> > > > We will incorporate this runtime analysis into the experimental evaluation of our revised manuscript.
> > > >
> > > > > W4: To my knowledge, ViT typically takes 224×224 images as input, whereas the authors’ method uses $\sqrt{T} < 224$ and therefore requires upsampling to 224, which introduces repeated pixels.
> > > >
> > > > Yes, the images are resized to the ViT input resolution. We refer the reviewer to Section 4.1 and Appendix D.1, where this step is analyzed in detail. Figure 10 illustrates how a very small or large patch size would result in many similar tokens. In contrast, choosing a patch size of sqrt(T) reduces the horizontal or vertical distortion and thus keeps redundancy low. While our nearest neighbor interpolation may introduce repeated adjacent pixels (ablation study on interpolation methods in Table 9), the overall patch/token diversity remains high due to our patch size heuristic.
> > > >
> > > > Note that the input transformation is only required to preserve the time series signal, but does not have to “introduce new information”, as suggested by the reviewer in their initial W4. Extracting meaningful features is the role of the ViT backbone.
> > > >
> > > > > Q1: Does the linear classifier mentioned by the authors require training? Where do its parameters come from?
> > > >
> > > > We refer the reviewer to Section 3 (l. 164) and Appendix C (l. 1056), where we detail our experimental setup: “we train a logistic regressor with the LBFGS solver per dataset”. Therefore, given ViT representations of dimensionality D (e.g. D=1280 for CLIP-ViT-H) and a dataset of C different classes, the number of parameters would be (D+1) x C. This is a common practice in TSFMs classification.
> > > >
> > > > As mentioned in our initial answer to Q1, we provide in Table 3 an additional evaluation with a nearest centroid classifier that does not require any learnable parameters.
> > > >
> > > > > Q4: Have the authors considered fusing channels before forming the final 3-channel representation, for example by using a convolution layer to aggregate information across channels?
> > > >
> > > > Fusing channels would require training an additional input projection per dataset. However, the main goal in our study is to keep the backbone frozen and avoid any training overhead except for the final classifier. This is a clear benefit over LLM-based methods for time series such as GPT4TS, which have to retrain an input embedding layer per dataset to bridge the modality gap.

---

> > > > > ### Comment · Reviewer_Rurf · 2025-11-19
> > > > >
> > > > > After reading the authors’ responses, I believe they have adequately addressed part of my earlier concerns. Their detailed and proactive clarifications have also improved my overall understanding of the work. As a result, I am inclined to slightly raise my rating of the paper.

---

> > > > > > ### Author Response · Authors · 2025-11-20
> > > > > > **All points were addressed. Which concerns remain?**
> > > > > >
> > > > > > We thank the reviewer for their continued engagement with our work and responses. We are pleased that **our clarifications have “improved [their] overall understanding of the work”** and led to an increased rating.
> > > > > >
> > > > > > We precisely addressed all questions raised by the reviewer. The following concerns should now be resolved:
> > > > > >
> > > > > > - Novelty (W1): We provided an extensive general comment on the concept of novelty in science and the specific novelty of our method. In our follow-up response, we **asked the reviewer six concrete questions for potential prior work** that might match our contribution. As the **reviewer did not name any such work** in their reply, we proceed with the understanding that **our submission differs significantly from prior work and is therefore novel**.
> > > > > > - Time series forecasting (W2): We referred the reviewer to our forecasting evaluation on 8 datasets in Section 4.6 (Table 22). We also clarified that **TiViT is designed specifically for discriminative tasks** such as classification due to the contrastive pretraining of its ViT backbone. **Forecasting-oriented TSFMs (VisionTS, Chronos) perform worse in time series classification.**
> > > > > > - Computational complexity and runtime (W3): We pointed the reviewer to our study of model parameters in Appendix D.6 (Table 12, Table 13). Moreover, we conducted an **additional quantitative runtime analysis** showing that TiViT with a ViT-B backbone is faster than VisionTS and close to Moment while achieving higher accuracy.
> > > > > > - Time series to image transformation: We clarified that the **patch size of sqrt(T) is a deliberate design choice to minimize the redundancy of patches** (W4, Q2), as detailed in Section 4.1 (Figure 3) and Appendix D.1 (Table 8, Figure 10). We further emphasized that our **transformation does not involve any learnable parameters**, which makes it more **lightweight** (Q3) than LLM-based TSFMs and allows for a separate modeling of channels in line with existing TSFMs (Q4).
> > > > > > - Classifier (Q1): We clarified that TiViT does not require any pretraining or end-to-end finetuning on time series and **explained the setup and parameter count of the linear classifier**. Since all methods are evaluated under the same protocol (logistic regression or nearest centroid classifier), the **performance improvements of TiViT can be attributed to our insightful leverage of the ViT backbone**.
> > > > > >
> > > > > > **The reviewer mentioned that we have “adequately addressed part of [their] earlier concerns”, but did not specify which issues remain. We are not aware of any unaddressed part.** Therefore, we would like to know:
> > > > > >
> > > > > > Which concerns remain that prevent our work from being accepted?
> > > > > >
> > > > > > We are eager to also address these. Thank you!

---

> > > > > > > ### Comment · Reviewer_Rurf · 2025-11-21
> > > > > > >
> > > > > > > I believe this paper is still some distance away from becoming a broadly convincing and insightful contribution. When discussing novelty, the authors repeatedly emphasize the superiority of empirical results, for example: “Our paper demonstrates that ViTs pretrained on natural images outperform existing TSFMs without any fine-tuning.” While such findings are useful, they are not, in my view, sufficient to establish true novelty.
> > > > > > >
> > > > > > > Most of the technical work appears to be combinations of existing engineering components—such as pairing ViT with OpenCLIP, integrating TiVIT with Mantis, or substituting different pretrained models like DINOv2/v3. I was unable to identify clear signs of the authors’ own algorithmic design or methodological innovation; much of the work feels mechanical and repetitive rather than conceptually new.
> > > > > > >
> > > > > > > That said, the authors did conduct an extensive set of experiments, and the empirical effort is non-trivial. Although the paper is marginally below my acceptance threshold, I would not strongly oppose acceptance if other reviewers find sufficient merit.
> > > > > > >
> > > > > > > If the authors have time, they may also elaborate on a few issues that are not central to the paper.
> > > > > > >
> > > > > > > 1. Clarification on “zero-shot” setting. The paper claims to be zero-shot, yet also acknowledges that “we train a logistic regressor with the LBFGS solver per dataset.” The boundary between zero-shot learning and task-specific adaptation needs to be clearly defined and justified.
> > > > > > > 2. Patch size choices. It may be beneficial to explore multiple patch sizes, especially in periodic or structured time series. In my experience, tuning patch granularity often leads to performance gains and offers additional insights.

---

> > > > > > > > ### Author Response · Authors · 2025-11-21
> > > > > > > > **Our methodological innovation is not the SOTA performance, but the steps we take and the analysis we conduct.**
> > > > > > > >
> > > > > > > > We are grateful to the reviewer for engaging in the discussion of our work. We appreciate that the **reviewer recognizes our “empirical effort is non-trivial”** and that the **reviewer “would not strongly oppose acceptance”**.
> > > > > > > >
> > > > > > > > We would like to apologize for any misunderstanding that our statement on the empirical superiority of our method may have caused. We do not claim that the superior performance by itself is the technical novelty of our paper. It only serves as an indicator that our approach is fundamentally different from all prior work.
> > > > > > > >
> > > > > > > > Instead, what we claim is that the **steps we had to take to achieve these superior results constitute the novelty of our work** (see Questions 3-6 in our second response). To be explicit:
> > > > > > > >
> > > > > > > > - No prior work has found that the **hidden representations** of pretrained ViTs are highly effective in time series classification. We examine the geometry of representations and **reveal a strong correlation between the intrinsic dimension/principal components of ViT layers and their downstream performance** (see Section 4.2, Figure 4, Figure 13).
> > > > > > > > - No prior work has **analyzed the representations of ViTs and TSFMs**, let alone recognized their **complementarity and consistent gains from fusing them**. We are the first to discover this complementarity and analyze the alignment/similarity of time series representations using the mutual kNN metric (see Section 4.3).
> > > > > > > > - No prior work has **systematically compared the 1D vs. 2D modeling of time series** from a theoretical and empirical point of view during pretraining (see Section 5).
> > > > > > > >
> > > > > > > > We believe these are **“insightful contributions”, not just “combinations of existing engineering components”**. Indeed, it wasn't obvious to us from the beginning that using hidden representations of frozen ViTs from pretrained vision-language models and fusing them with representations of TSFMs would be so impactful for time series classification. But to our surprise (and also to the surprise of reviewer vtvJ) it is. We believe that our findings are important to be shared for others not to have to discover the same technical improvements or to conduct the same representational analysis, and also for practitioners to benefit from our principled approach.
> > > > > > > >
> > > > > > > > Furthermore, we would like to use the opportunity to elaborate on the final two issues raised by the reviewer:
> > > > > > > >
> > > > > > > > - Zero-shot setting: We acknowledge that our current setup does not support zero-shot classification in the sense of CLIP. This would require an additional alignment between the representations of time series and textual labels.
> > > > > > > > However, in the time series domain, recent work has often referred to our setting as “zero-shot protocol” [1] or “zero-shot feature extraction” [2]. The closest we can get to zero-shot is using a nearest-centroid classifier without learnable parameters on top of our frozen backbone (see Section 4.6).
> > > > > > > > We agree with the reviewer that the time series community should clearly define the boundary between zero-shot learning and task-specific adaptation. **To be precise and avoid ambiguity, would the reviewer agree with us if we consistently use the term “zero-shot feature extraction”?**
> > > > > > > > - Patch size choices: We perform a **hyperparameter search for the optimal patch size** P* **and patch overlap** on each of the 128 UCR datasets (see Section 4.1 and Appendix D.1). For each dataset, we consider 20 different patch sizes in $[1, T/2]$, and 3 levels of overlap in $[0, 0.5 P, 0.9 P]$.
> > > > > > > > An exhaustive search for P* offers marginal improvements over $P=\sqrt{T}$ in the case of no overlap. However, **introducing overlap between patches boosts performance and makes the impact of the optimal patch size vanish** (see Figure 3). Therefore, we recommend using a patch size of $P=\sqrt{T}$ and a stride of $S= P/10$.
> > > > > > > > Note that this is another **non-trivial finding that distinguishes our method from previous vision-based approaches for time series**. For instance, VisionTS selects patch sizes based on known sampling frequencies of forecasting datasets and proposes $P=1$ when time series lack clear periodicity. We show that this does not transfer to time series classification.
> > > > > > > >
> > > > > > > > The novelty of our work resides in our methodological steps and analytical insights, not merely the SOTA performance. **We hope that our clarifications will resolve the concerns of the reviewer that kept our paper “marginally below [their] acceptance threshold”.**
> > > > > > > >
> > > > > > > > [1] Auer et al. "Pre-trained Forecasting Models: Strong Zero-Shot Feature Extractors for Time Series Classification." 2025.
> > > > > > > > [2] Feofanov et al. "Mantis: Lightweight Calibrated Foundation Model for User-Friendly Time Series Classification." 2025.

---

> > > > > > > > > ### Comment · Reviewer_Rurf · 2025-11-21
> > > > > > > > >
> > > > > > > > > Thank you for the authors’ detailed response. I still hold my original view regarding the paper. I encourage the authors, together with the other reviewers, to further discuss and explore possible directions for improvement.

---

### Official Review · Reviewer_o3Rw · 2025-10-31

**Soundness:** 2
**Presentation:** 2
**Contribution:** 2
**Rating:** 2
**Confidence:** 3

**Summary:**

The paper introduces Time Vision Transformer (TiViT), a new approach to time series classification by leveraging pretrained Vision Transformers (ViTs). TiViT transforms time series data into images and utilizes the powerful representations from these pretrained models, achieving state-of-the-art performance in both time series classification and anomaly detection.

**Strengths:**

The approach transforms time series into 2D images and uses pretrained, frozen ViTs for feature extraction, without the need for training or fine-tuning on individual datasets. The method is straightforward and reusable across different downstream tasks.

**Weaknesses:**

1. This paper presents an innovative framework and demonstrates its superior performance through experiments. However, there is room for improvement in the comparison with existing methods and the depth of theoretical analysis. I recommend the authors further clarify TiViT's innovative points, explain the differences from related works (such as NuTime, VisionTS), and strengthen the discussion of the advantages of the 2D conversion method.

2.  The paper does not provide a detailed explanation of why the frozen image-based ViT is effective in extracting time series features such as periodicity, trends, and non-stationarity. What is the relationship between these different modalities (image and time series ) of input?

**Questions:**

Q1: While the idea of converting time series to images and using pretrained vision models (like ViT) for classification is interesting, it is not entirely novel. Many similar approaches, such as VisionTS, already exist. The authors should clarify TiViT's unique advantages and innovative aspects over existing methods.

Q2: The approach in TiViT seems very similar to VisionTS, except for using ViT instead of MAE. The authors should elaborate on how TiViT differs from VisionTS and explain why ViT is a better choice for this task.

Q3: The authors mention that “2D representations are more suitable than 1D for time series tasks” needs further exploration. I suggest the authors enhance the ablation studies in section 4.5, with a deeper comparison of 2D vs. 1D representations across different datasets.

Q4: The 2D conversion method is a key innovation. The authors should expand on the advantages of 2D conversion and compare it to
other methods like line plots and Gramian angular fields to clearly highlight its benefits. This will help to clearly demonstrate the practical benefits of 2D conversion and its impact on model performance.

Q5: Reference update:  the paper cites VisionTS: Visual Masked Autoencoders, which has now been accepted by ICML 2025, and should be updated in the references lists.

---

> ### Author Response · Authors · 2025-11-17
>
> We thank the reviewer for acknowledging our “innovative framework” and “its superior performance” on time series classification and anomaly detection benchmarks.
>
> > Q1/W1. “The authors should clarify TiViT's unique advantages and innovative aspects over existing methods.”
>
> We refer the reviewer to our general comment on novelty. The limitations of existing methods modeling time series as images are outlined in Section 1 and 2 of our paper:
> - TimesNet (Wu et al., 2023) has been trained end-to-end per dataset on heatmaps generated from time series. This fully supervised approach only reaches 57.2% on the UCR benchmark (see Figure 5).
> - VisionTS (Chen et al., 2024) has applied pretrained MAEs in time series forecasting. Adopting this approach to time series classification only reaches 73.4% (see Figure 5).
> - Li et al. (2023b) have finetuned SwinTransformer on line plots of irregularly sampled time series. Their approach has only been evaluated on a very specific task, while we evaluate TiViT on the most common time series benchmarks.
> - NuTime (Lin et al., 2023) does not model time series as images. Feofanov et al. (2025) have shown that NuTime is inferior to Mantis in time series classification, which we outperform in our work.
>
> > Q2/W1. “The approach in TiViT seems very similar to VisionTS, except for using ViT instead of MAE. The authors should elaborate on how TiViT differs from VisionTS.”
>
> First, we would like to remind the reviewer that MAE is a pretraining method for ViTs. In fact, both TiViT and VisionTS incorporate the ViT architecture. However, there are several other crucial differences between both methods as mentioned in Section 1 (l. 48), Section 2 (l. 130), Section 4.2 (l. 267), and Appendix D.9 (l. 1329):
> - VisionTS focuses on the task of forecasting, while we primarily investigate classification and anomaly detection.
> - VisionTS employs a ViT backbone pretrained with the MAE framework for forecasting. Figure 5 shows that adopting this approach in time series classification yields only 73.4%, which is far from the state-of-the-art. Our key insight is that the most effective representations for classification are hidden within the ViT.
> - VisionTS is limited to ViTs from MAEs, while we investigate ViTs from more recent vision and vision-language models such as DINOv3, CLIP, and SigLIP 2.
>
> > Q3. “The authors mention that “2D representations are more suitable than 1D for time series tasks” needs further exploration. I suggest the authors enhance the ablation studies in section 4.5, with a deeper comparison of 2D vs. 1D representations across different datasets.”
>
> We would like to point out that this direct quote does not appear anywhere in our paper.
> The ablation studies in Section 4.5 are conducted on 128 UCR datasets. Furthermore, our paper includes extensive comparisons of 2D vs. 1D representations across different datasets. In Section 4.2, we compare the effectiveness of 2D vs. 1D representations in time series classification on 128 univariate and 27 multivariate datasets (see Table 1). In Section 4.3, we compare the similarity of 2D vs. 1D representations using the mutual k-NN metric (see Table 2) on the 10 largest UCR datasets. In Section 4.4, we provide a qualitative comparison of 2D vs. 1D representations (see Figure 6). Finally, in Section 5.2, we perform pretraining with 2D vs. 1D representations on 8 datasets and evaluate the representations on 128 UCR datasets.
>
> > Q4. “The authors should expand on the advantages of 2D conversion and compare it to other methods like line plots and Gramian angular fields to clearly highlight its benefits.”
>
> We would like to refer the reviewer to Appendix D.3, where we compare TiViT with different imaging methods including line plots and GAFs. TiViT achieves the highest classification accuracy using our heatmap-based representations.
>
> > Q5. “the paper cites VisionTS [...] which has now been accepted by ICML 2025, and should be updated in the references lists.”
>
> We thank the reviewer for this hint and will update the reference.
>
> > W2: “The paper does not provide a detailed explanation of why the frozen image-based ViT is effective in extracting time series features [...] What is the relationship between these different modalities (image and time series ) of input?”
>
> TSFMs are trained on millions of time series, while foundation models in the vision domain are trained on billions of natural images. This large-scale pre-training enables ViTs to learn discriminative features. In Appendix D.11, we refer to the work of Ghiasi et al. (2022) showing that earlier ViT layers capture general edges and textures. Such features are also present in images generated from time series.
> In Section 4.4, we provide qualitative results with attention rollout and t-SNE visualization to gain insights into the processing of TiViT. In Section 4.5, we further study the relationship, in particular the similarity, of image and time series representations using mutual k-NN.

---

> > ### Author Response · Authors · 2025-11-24
> > **Follow-up**
> >
> > Dear Reviewer,
> >
> > Thank you once more for the constructive feedback you have given regarding our submission. We are writing to ask if you have had a chance to consider our previous response and our general comment regarding the novelty of our work. We hope that our clarifications address the concerns raised in your initial review. If any issues remain, we are ready to have a discussion and provide more information regarding them.
> >
> > As reviewer Rurf indicated, we hope that a consensus can be reached regarding this paper and your input will be invaluable to reach such a consensus. Thank you in advance for all your help in making our submission stronger.

---

> > > ### Comment · Reviewer_o3Rw · 2025-11-25
> > >
> > > Thank you for the authors’ detailed response. I believe the entire framework is exactly as the paper described, combining two methods (TiViT + TSFMs), and achieving good results in experiments. However, the deeper principles and explanations behind these method might be what can provide readers with better insights. Therefore, I still think that if the authors can think further about this issue, it will lead to better discoveries.

---

> > > > ### Author Response · Authors · 2025-11-26
> > > > **Section 4 and 5 provide “deeper principles and explanations”**
> > > >
> > > > We thank the reviewer for their response. **We fully agree that “the deeper principles and explanations behind [our] method might be what can provide readers with better insights”. That is why we analyze the representations of ViTs and TSFMs in Section 4.2, Section 4.3, and Section 4.4 of our paper.**
> > > >
> > > > As highlighted in our previous response and in our general comment, the novelty of our method is not merely the SOTA performance in time series classification, but all the steps we had to take and analyses we had to conduct in order to achieve these superior results.
> > > >
> > > > - In **Section 4.1**, we **examine the impact of patch size and overlap** through a hyperparameter search. We show that selecting a patch size of sqrt(T) and a patch overlap of 0.9 eliminates the need to tune the patch size per dataset (see Figure 3) and reduces token redundancy (see Figure 10).
> > > > - In **Section 4.2**, we **investigate the intrinsic dimension of time series representations across ViT layers** (see Figure 4) and reveal a **strong correlation between the intrinsic dimension and classification performance** (Pearson correlation of 0.7). This representational insight is the key to our superior performance and differentiates our work from any other vision-based approach for time series. Note that simply using final-layer representations of ViTs remains far from SOTA results on the UCR benchmark (see Figure 4a and VisionTS in Figure 5).
> > > > - In **Section 4.3**, we **study the representations of ViTs and TSFMs using the mutual kNN metric** to quantify their (dis-)similarity. We are the first to reveal that there is a slight alignment of TSFMs and TiViT, but given the score between 0.2 and 0.3, their local representations are still fairly different and thus contribute complementary information to the SOTA joint classification.
> > > > - In **Section 4.4**, we provide a **qualitative comparison of TiViT and TSFMs**. Specifically, we employ **attention rollout** (Figure 6a, Figure 17) and illustrate that TiViT attends to salient regions of the time series images. **t-SNE visualizations** (Figure 6b, Figure 16) show that, without any training on time series, TiViT generates embeddings that form clusters aligned with the ground-truth classes.
> > > > - In **Section 5**, we provide theoretical and empirical **evidence for the benefits of 2D modeling over 1D modeling in time series classification**. As illustrated in Figure 7 and Figure 8, 2D modeling can better spread label-relevant information across tokens and thus facilitates time series classification. We further show that the contrastive pretraining of a Transformer model with 2D patching yields better downstream performance than with 1D patching (see Table 5).
> > > >
> > > > We would like to emphasize that all these analyses are already part of our paper, and we believe that they actually address the reviewer’s request for “deeper principles and explanations.”
> > > >
> > > > **Unfortunately, the reviewer’s comment remains rather general and does not engage with our detailed, point-by-point response. Therefore, it is unclear what “better insights” or “better discoveries” the reviewer still expects. We remain eager to provide further clarifications, but can only do so if there is a specific request regarding our work or previous responses. Otherwise, we have to assume that our rebuttal has precisely addressed every point of the reviewer.**

---

### Official Review · Reviewer_vtvJ · 2025-11-01

**Soundness:** 2
**Presentation:** 3
**Contribution:** 1
**Rating:** 4
**Confidence:** 3

**Summary:**

The paper titled “TiViT: Time Series Representations Lie Hidden in Pretrained Vision Transformers
 ”introduces Time Vision Transformer (TiViT), a framework that leverages the pre-trained frozen Vision Transformers (ViTs), like OpenCLIP, by converting time series data into images for classification and anomaly detection. TiViT significantly outperforms conventional Time Series Foundation Models (TSFMs) on time series classification benchmarks. The success is attributed to the inherent advantages of 2D patching for time series modeling, which is shown both theoretically and empirically to increase the proportion of "label-relevant tokens" and make learning more efficient than traditional 1D modeling.

**Strengths:**

The proposed model seems to be working well given that it has been shown to achieve state-of-the-art performance on time series classification and anomaly detection benchmarks.

Its surprising to see  that large, frozen Vision Transformers (ViTs) pre-trained solely on natural images can serve as universal feature extractors for non-visual domains like time series analysis.

**Weaknesses:**

There is not much in terms of novelty in the paper. The overall process of converting time series into images to leverage visual representations is acknowledged as an existing approach in the field. The other changes seem pretty straightforward.

The theoretical insight regarding the benefit of 2D patching relies on a framework originally introduced for shallow Vision Transformers on generic data. The theoretical proposition is constructive and based on specific assumptions about pattern placement, which may not hold generally.

While the authors compare against state-of-the-art TSFMs like Mantis and Moment, the broader comparison to the vast array of existing time series classification methods (especially modern, non-FM approaches trained from scratch) is limited to citing results from a previous work (Goswami et al. (2024)).

**Questions:**

na

---

> ### Author Response · Authors · 2025-11-17
>
> We appreciate that the reviewer recognizes our “state-of-the-art performance on time-series classification and anomaly detection benchmarks.” Furthermore, we are pleased to read that they find it “surprising” that ViTs serve as universal feature extractors for time series tasks. That being said, we struggle to reconcile the reviewer's acknowledgement of our surprising findings with the reviewer's claim that our work lacks novelty (W1). We address this contradiction in our general comment on novelty and provide further clarifications below.
>
> > W2: “The theoretical proposition is constructive and based on specific assumptions about pattern placement, which may not hold generally.”
>
> Our theoretical insights show that the 2D patching of time series can increase the fraction of label-relevant tokens over 1D patching, and thus facilitates classification. We validate this insight with two experiments on real-world time series.
>
> In Section 5.1, we demonstrate on samples from the UCR repository that 2D modeling spreads label-relevant information. We train shallow ViTs with 1D and 2D modeling on the BirdChicken dataset and utilize MILLET to obtain interpretability scores for each timestamp within a time series. Figure 8 shows that the interpretability heatmaps for 2D samples carry fewer signals for the wrong class than the heatmaps for 1D samples.
>
> In Section 5.2, we compare pretraining a Transformer on real-world data with 1D patching vs 2D patching, where the Transformer architecture, the pre-training dataset, and the contrastive pre-training method are all fixed, and only the patching strategy is different. We then evaluate the pretrained Transformer models on the UCR benchmark and find that the model pretrained with 2D patching reaches a higher downstream accuracy than the model pretrained with 1D patching.
>
> > W3: “the broader comparison to the vast array of existing time series classification methods (especially modern, non-FM approaches trained from scratch) is limited to citing results from a previous work”
>
> We benchmark TiViT against 19 existing methods in Figure 5. We evaluate the TSFMs Mantis and Moment on our own since they are the most related to our work and an integral part of our study on feature complementarity and joint classification (see Section 4.3). We further show that TSFMs such as Chronos and VisionTS, primarily designed for time series forecasting, perform much worse than TiViT in time series classification. To the best of our knowledge, we are the first to make this comparison (see Section 4.2).
> For a full comparison against self-supervised and supervised baselines, we do cite results from Goswami et al. (2024). Their paper is a fundamental work in the area of TSFMs and reports the classification performance of models on the same UCR benchmark. Citing results from published work is not only a standard practice, but ensures consistency and a fair comparison.

---

> > ### Author Response · Authors · 2025-11-24
> > **Follow-up**
> >
> > Dear Reviewer,
> >
> > Thank you once more for the constructive feedback you have given regarding our submission. We are writing to ask if you have had a chance to consider our previous response and our general comment regarding the novelty of our work. We hope that our clarifications address the concerns raised in your initial review. If any issues remain, we are ready to have a discussion and provide more information regarding them.
> >
> > As reviewer Rurf indicated, we hope that a consensus can be reached regarding this paper and your input will be invaluable to reach such a consensus. Thank you in advance for all your help in making our submission stronger.

---

### Author Response · Authors · 2025-11-17
**Novelty**

What is novelty in science? Michael J. Black, Director of the Max Planck Institute for Intelligent Systems, offers an answer to this question in a blog post [1]:

**“The simplicity of an idea is often confused with a lack of novelty when exactly the opposite is often true. [...] The inventive insight is to realize that a small change could have a big effect.”**

We encourage the reviewers to read his entire post.

Our paper demonstrates that ViTs pretrained on natural images outperform existing TSFMs without any fine-tuning. **The previous vision-based approaches for time series modeling (VisionTS, TimesNet) are not even close to the performance of our method TiViT in time series classification.** VisionTS, cited by several reviewers, is ranked 29th on the GIFT-Eval leaderboard of forecasting models. Its empirical performance on the forecasting task cannot be compared to TiViT’s SOTA performance in classification. **Selecting the hidden representation of ViTs with the highest intrinsic dimension is a crucial and non-trivial design choice to achieve these results.** It is an **“inventive insight” under Black’s definition of novelty**. In Section 4.2, we provide an in-depth investigation into the geometry (see Figure 4b) and effectiveness (see Figure 4a) of hidden ViT representations for time series classification.

Simply using final-layer representations of ViTs in time series classification remains far from SOTA results on the UCR benchmark. Figure 5 shows that **VisionTS** with an MAE backbone and final-layer representations **only reaches 73.4%** accuracy. **TimesNet**, another vision-based and fully supervised approach, is even more inferior and **only reaches 57.2%**. In contrast, our method **TiViT achieves 83.5%** and the joint classification of TiViT and Mantis even yields 84.8%. Assuming our method was not novel, as the reviewers claim, could they please explain how we nonetheless outperform all prior vision-based models and even the most recent time series foundation models Mantis and Moment by a significant margin?

Following Black’s definition, novelty arises from results that are both surprising and high-impact. **All three reviewers recognize the SOTA performance of TiViT** on time series classification and anomaly detection benchmarks. In addition, **our findings are described as “inspiring” (reviewer Rurf) and “surprising” (reviewer vtvJ)**. These reactions demonstrate that TiViT produces outcomes that reviewers did not anticipate, which would by itself satisfy the criterion of novelty. **We respectfully ask the reviewers to weigh this evidence when assessing the novelty of our contribution.**

Additionally, we present at least two **more findings** that, to the best of our knowledge, have not appeared in any prior work:
- We are the first to reveal that **ViTs and conventional TSFMs learn complementary features** that can be combined to achieve even higher accuracy (see Section 4.3, Table 2). The **joint classification** of TiViT and Mantis yields **performance gains of +3%** on the UCR benchmark over the TSFM Mantis.
- We provide the first theoretical justification for 2D time series modeling and empirically show that **pretraining with 2D patching leads to better downstream performance than 1D patching** (see Section 5).

[1] Michael J. Black. “Novelty in Science. A guide to reviewers.” 2022.

---

### Meta-Review · Area_Chair_GZtN · 2026-01-06

**Summary:**

This manuscript introduces TiViT, a framework that repurposes frozen Vision Transformers pretrained on large-scale image data for time-series classification and anomaly detection by mapping temporal signals into two-dimensional representations. The paper reports strong empirical performance across widely used benchmarks and provides an extensive representational analysis, including intrinsic dimensionality, layer-wise effectiveness, and complementarity with existing time-series foundation models.

Reviewers broadly agreed that the experimental evaluation is thorough and that several empirical findings are interesting, particularly the observation that intermediate ViT representations transfer effectively to time-series tasks. However, there was consistent concern regarding conceptual novelty and methodological distinctiveness. The main ingredients of the approach—time-series-to-image conversion, reliance on frozen pretrained ViTs, and linear probing—have been explored in prior work, and the contribution was largely perceived as an incremental recombination of existing ideas rather than the introduction of a clearly new modeling principle.

The rebuttal provided detailed clarifications and additional analyses, including runtime measurements and further representational studies. While these additions improved clarity, they did not fully resolve the central concern that the contribution is primarily empirical and analytical. Reviewers remained unconvinced that the paper establishes a novel algorithmic or theoretical insight beyond demonstrating the effectiveness of transferring vision representations to time-series classification. In particular, stronger differentiation from prior vision-based time-series methods, clearer isolation of what is uniquely enabled by the proposed design choices, and a more explicit framing of the representational analyses as general principles rather than post-hoc observations would be necessary to elevate the contribution.

In summary, the paper presents solid empirical results and careful analysis, but it does not yet meet the bar for a sufficiently distinct or transformative contribution. With sharper conceptual framing and clearer methodological innovation, the work could form the basis of a strong future submission.

**Reviewer Concerns:**

Reviewer vtvJ: Acknowledged the strong empirical performance of pretrained vision models on time-series tasks but rated the contribution low in novelty, noting that time-series-to-image conversion and ViT-based classification have been explored previously. They questioned the generality of the theoretical arguments and the breadth of baseline comparisons. These concerns were not fully resolved in the rebuttal.

Reviewer o3Rw: Recognized the promise of the approach and its empirical results but emphasized that key components (time-series-to-image transformation and pretrained vision backbones) are not new. They requested deeper explanation of why frozen ViT representations capture time-series structure and clearer differentiation from related work such as VisionTS. The rebuttal largely reiterated existing analyses without fully addressing these points.

Reviewer Rurf: Expressed continued reservations about novelty, viewing the method as a recombination of existing components despite extensive experiments. They noted limitations in scope and practical considerations, and maintained that strong empirical performance alone does not constitute a novel methodological contribution.

**Reviewer Scores:**

My expectation is that none of the reviewers would increase their score, given that their primary concerns persisted after the rebuttal.

---

### Decision · Program_Chairs · 2026-01-26

Reject